# *Helicobacter pylori* initiates successful gastric colonization by utilizing L-lactate to promote complement resistance

Shuai Hu [1] & Karen M. Ottemann [1] ✉

The complement system has long been appreciated for its role in bloodborne infections, but its activities in other places, including the gastrointestinal tract, remain elusive. Here, we report that complement restricts gastric infection by the pathogen *Helicobacter pylori*. This bacterium colonized complement-deficient mice to higher levels than wild-type counterparts, particularly in the gastric corpus region. *H. pylori* uses uptake of the host molecule L-lactate to create a complement-resistant state that relies on blocking the deposition of the active complement C4b component on *H. pylori*'s surface. *H. pylori* mutants unable to achieve this complement-resistant state have a significant mouse colonization defect that is largely corrected by mutational removal of complement. This work highlights a previously unknown role for complement in the stomach, and has revealed an unrecognized mechanism for microbial-derived complement resistance.

Animals have multilayered mechanisms to combat pathogen exposure. The gastrointestinal (GI) tract in particular experiences regular exposure to numerous microbial species, and therefore employs multiple strategies to restrict colonization by the pathogens in this mix. These restrictions include mechanical barriers, competition from other microbes, and innate immune mechanisms. One critical non-cellular innate immune mechanism is the complement system. It remains unknown whether this system plays vital defensive role in the GI tract and whether GI microbes have distinct ways to deal with this challenge.

The complement system plays crucial roles in defending microbial infection by inducing numerous immune responses, such as direct target cell lysis through formation of membrane attack complex (MAC); recruitment, activation, and regulation of phagocytic cells via anaphylatoxins release; and modulation of adaptive immune responses[1–4]. Complement components are a set of proteins that are mainly produced in the liver, and released into the blood where they travel to tissues throughout the body. Because of its prevalence in the blood, complement is well known to play critical roles in bloodborne defenses. The function of the complement system in GI tract defense, in comparison, has been relatively understudied, partially because only a few complement components are detectable in the GI lumen of healthy hosts[5]. Complement components have been detected in the GI tract during chronic diseases, including Crohn's disease, ulcerative colitis, and chronic gastritis[6–8]. These findings suggest complement proteins are present in the GI tract and may play unappreciated roles.

The complement system is set in action by one of three triggering pathways, called classical, lectin, or alternative. Each one is triggered by a different molecule, e.g. classical is triggered by the Fc portion of antibodies protruding from the targeted pathogen, while lectin is triggered by specific surface carbohydrates. After the complement system is activated, it sets off a cascade of proteolytic reactions that process pre-proteins to their active forms (Fig. 1a). In the classical and lectin pathways, a key step is the processing of C4 to its products C4a and C4b, which complex with activated C2 to catalyze the formation of C3 convertase. C3 convertase formation is a central step in the alternative pathway as well. Multiple proteolytic steps follow this central step to eventually result in activated products and MAC formation.

Many microbes infect the GI tract, suggesting they have developed sophisticated strategies to overcome innate immune restrictions. One pathogen that is well known for its ability to establish and maintain GI colonization is *Helicobacter pylori*. *H. pylori* infects the stomachs of millions of people worldwide and is a leading cause of gastritis, peptic ulcer disease, mucosa-associated lymphoid tissue

[1]Department of Microbiology and Environmental Toxicology, University of California, Santa Cruz, CA 95064, USA. ✉e-mail: ottemann@ucsc.edu

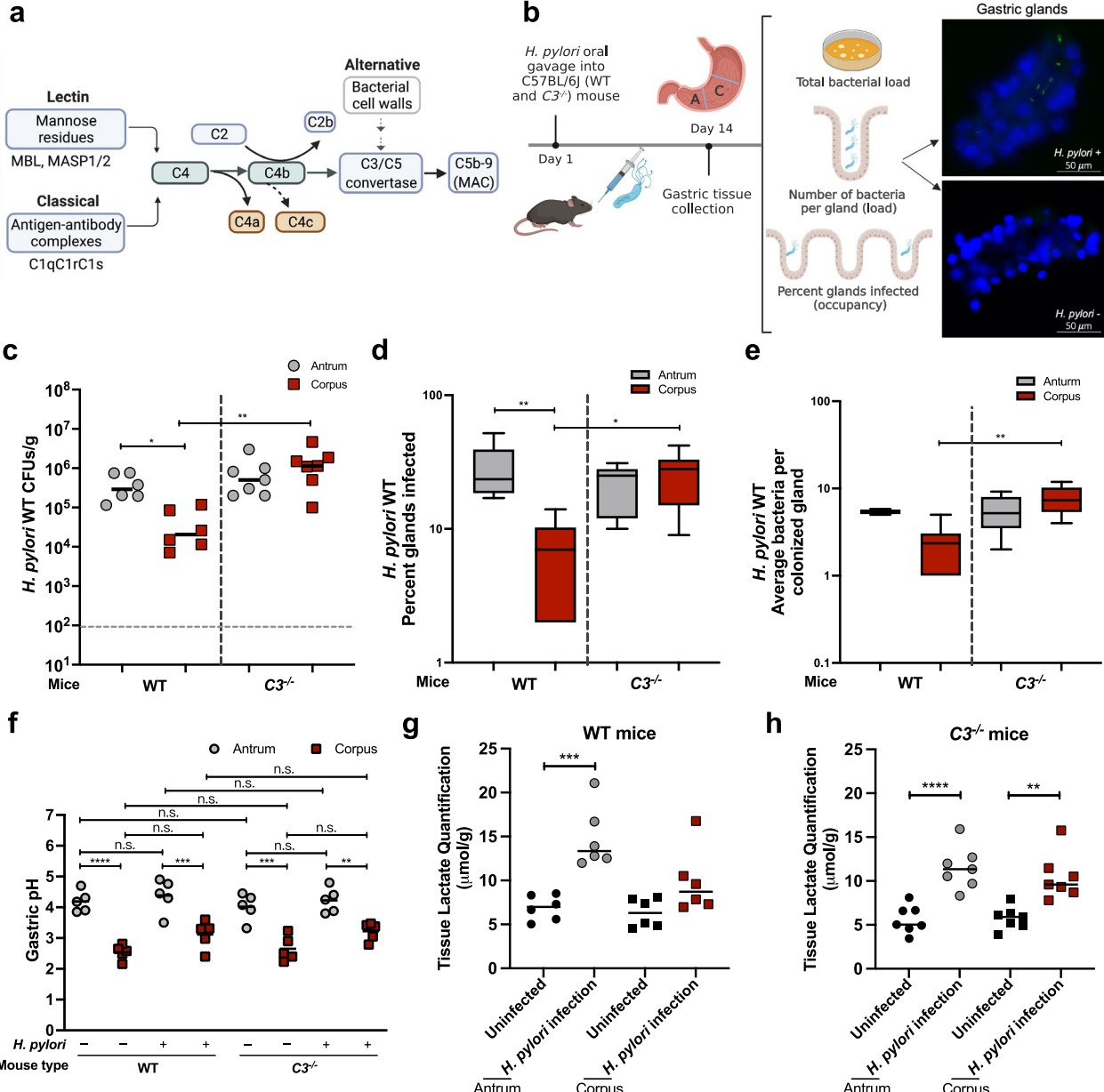

**Fig. 1 | Complement plays a vital role in restricting *H. pylori* gastric colonization. a** Schematic diagram of complement pathways activation. **b** Schematic illustration of mouse infection analysis. C57BL/6J WT ($n = 6$) or $C3^{-/-}$ ($n = 7$) mice were infected by oral gavage with GFP⁺ *H. pylori* PMSS1 for two weeks. Stomachs were isolated from infected mice and separated into corpus and antrum pieces and processed to obtain the total bacterial numbers (colony forming units, CFUs) by plating, to enumerate bacteria in glands using microscopy. Bacteria were visualized by GFP expression (green), while gland cells were visualized using a DNA Hoechst stain (blue). **c** Total bacterial number from gastric antrum (gray) and corpus (red) of infected WT or $C3^{-/-}$ mice normalized to tissue weight (CFUs/g). Gray dotted line, limit of detection. **d** Percentage of glands occupied by *H. pylori* in the gastric antrum (gray) and corpus (red) of infected WT or $C3^{-/-}$ mice, determined by analyzing 100 randomly selected isolated glands of each infected mouse. **e** Number of *H. pylori* per colonized gland in the gastric antrum (gray) and corpus (red) of infected WT or $C3^{-/-}$ mice. **f** Gastric pH levels in either antrum or corpus region of both WT and $C3^{-/-}$ mice, in *H. pylori* free or infected condition. **g**, **h** Gastric L-lactate levels were measured using samples for bacterial plating. For all dot plots, each dot represents one mouse sample. For all panels, black bars represent mean of samples. For plot d and e, the box shows the 10th to 90th percentiles of the dataset; the black whiskers represent the maxima and minima of the dataset. The *p* values were obtained by one-way ANOVA with Tukey's multiple comparisons test (**c**, **d**, **e**, **f**), and unpaired two-tailed *t*-tests (**g**, **h**). The significance is indicated as * ($p < 0.05$); ** ($p < 0.01$). Source data and exact *p* values are provided in the Source Data file.

(MALT), and gastric adenocarcinoma[9–11]. Previous observations have suggested there is an interaction between *H. pylori* and human complement. For example, *H. pylori* triggers activation of human complement in vitro[12]. Furthermore, *H. pylori*-infected patients exhibit significantly higher levels of the C3 complement protein in the gastric samples as compared to *H. pylori* negative ones[8]. Two additional studies suggest *H. pylori* can thwart the action of complement using host proteins, CD59 and vitronectin, that normally protect host cells from complement destruction[13,14]. These known *H. pylori* resistance mechanisms rely on recruiting host proteins, but there are not any known *H. pylori* self-produced complement evasion mechanisms unlike found in other microbial species, such as *Pseudomonas* spp. and *Staphylococcus* spp. that produce various proteases to inactivate complement components[15]. Thus, the relevance between *H. pylori*

infection and gastric complement activation has been suggested but the features of these interactions have not yet been elucidated.

Many pathogens sense aspects of the host environment and upregulate defensive mechanisms[16–18]. In the case of complement resistance, it is well known that the host molecule L-lactate promotes complement resistance in *Neisseria meningitidis* by activating a sialic acid biosynthesis pathway that protects against complement's action[19]. We were intrigued by this response, because L-lactate is known to be important for *H. pylori*. *H. pylori* senses L-lactate as a chemoattractant, directing its migration toward this signal[20]. In addition, L-lactate is one of the most depleted chemicals when *H. pylori* grows with gastric organoids, and can be taken up and used for growth[21–23]. Given lactate's importance, we asked whether it similarly stimulates an anti-complement mechanism in *H. pylori*.

In this study, we explored the importance of the complement system in gastric defense and the mechanisms used by *H. pylori* to avoid its action. We discovered that the complement system restricts *H. pylori* proliferation in the murine stomach, in a region specific manner. *H. pylori* employs L-lactate, in turn, to create a bacterial state that inhibits activation of the classical complement pathway; L-lactate uptake is critical for this response and required in vivo. Finally, we uncovered that this resistance is accomplished in a previously undocumented manner by destabilizing the interaction between the C4b product and the *H. pylori* surface. Overall, our results reveal that complement plays an important role in controlling bacterial colonization in the gastric region of GI tract, with GI tract bacteria utilizing previously undescribed mechanisms to overcome this challenge.

## Results

### Complement is important in gastric defense
Our first goal was to characterize the role of complement in bacterial stomach colonization. We made use of mice that lack the critical complement C3 protein ($C3^{-/-}$) (Fig. 1b). Mice lacking C3 are unable to form MAC and induce cell lysis that results from any of the triggering pathways[24] (Fig. 1a), and less capable of recruiting immune cells to the site of infection[2]. $C3^{-/-}$ and WT control C57BL/6 J mice were infected with the commonly used mouse-adapted *H. pylori* strain PMSS1[25]. After two weeks, total bacterial counts and gland colonization were evaluated in each of the two main stomach regions, antrum and corpus. As reported by others, *H. pylori* colonized the WT antrum to higher levels than the corpus[26,27]. Total bacterial load in the $C3^{-/-}$ mice corpus was significantly elevated 100-fold compared to WT mice (Fig. 1c), demonstrating some of the restrictions were removed in the $C3^{-/-}$ mice. Colonization in the important gastric subregion, the gastric glands was quantified as exhibited in Fig. 1b, also showed that complement affected colonization most significantly in the corpus. The percent of infected glands and number of bacteria per gland each increased by fourfold (Fig. 1d, e). As gastric acid, mainly produced in the corpus, is an important antimicrobial host factors in the GI tract, we examined whether $C3^{-/-}$ mice have differences in gastric pH compared to WT mice and detected no differences between mouse strains, or with and without infection (Fig. 1f). These results suggest complement restricts *H. pylori* gastric colonization, especially in the corpus region.

### *H. pylori* is sensitive to complement-mediated elimination
The above results show that removing complement from the mouse model elevates *H. pylori* numbers. Complement can both mediate direct cell lysis and facilitate local immune response. *H. pylori* infections recruit immune cells slowly, and studies have shown that during the infection period used here (14 days), there is limited immune cell recruitment to the site of *H. pylori* infection[28], suggesting that a main role for complement may be to act via direct cell lysis. We therefore examined whether this *H. pylori* strain is vulnerable to

complement-mediated cell lysis, as had been seen for other strains[12], using exposure to active normal human serum (NHS). NHS was utilized here as it contains abundant complement components and is able to be activated without *H. pylori* specific antibodies[12]. Active NHS eliminated *H. pylori* in a dose-dependent manner, while heat-treated inactive NHS was unable to kill *H. pylori* (Fig. 2a). Treatment with 10% NHS eliminated 60% of *H. pylori*, while treatment with 30% NHS eliminated 85% (Fig. 2a). Considering that *H. pylori* colonizes the stomach where it is acidic, we examined whether complement retains function in a low pH environment. We exposed *H. pylori* to 10% or 30% NHS at pH 3.5, and found that the complement was still active to significantly decrease *H. pylori* numbers, killing 40% or 92% of *H. pylori*, respectively (Fig. 2b). These results suggest that complement retains at least some killing ability at low pH. To verify whether this elimination was complement-activation dependent, we treated the NHS with EGTA, a compound that blocks all three complement activation pathways[12,29,30]. EGTA-treated NHS was unable to eliminate *H. pylori*, consistent with the idea that the active component in NHS is complement (Fig. 2c).

To validate whether this finding is applicable to the murine model and fully complement dependent, we extracted mouse serum from *H. pylori*-infection-free WT and $C3^{-/-}$ mice, and examined *H. pylori* survival ratio in each type of mouse serum. WT mouse serum eliminated around 75% of bacteria, while $C3^{-/-}$ mouse serum was only able to eliminate about 10% of the *H. pylori*, leaving nearly 90% of the population to survive (Fig. 2d). These results strongly suggest that complement activation is a major cause of *H. pylori* elimination during serum treatment.

### Complement-mediated clearance depends on classical pathway activation
Complement can be activated through three pathways, classical, lectin, or alternative. These pathways can be blocked by treating with EGTA, and then differentially re-activated by adding back either $Ca^{2+}$, to activate the classical and lectin pathways, or $Mg^{2+}$, to activate the alternative pathway. We used this differentially-activated serum to evaluate which pathway targeted *H. pylori*. The presence of EGTA completely abrogated the susceptibility of *H. pylori* to complement killing (Fig. 2c). $Ca^{2+}$ supplementation resulted in the killing of *H. pylori* to levels that were nearly identical to those in untreated NHS, suggesting that *H. pylori* is mostly targeted by the classical and/or lectin pathways (Fig. 2c). This observation was confirmed using Factor-B-depleted serum, which loses the alternative pathway (Fig. 2c). To further distinguish the role of the classical from the lectin pathway, C1q-depleted serum was applied to exclusively block classical pathway activation[2], and had no obvious effect on bacterial viability, suggesting that the classical pathway played a critical role while the lectin pathway played a minor role in *H. pylori* elimination (Fig. 2c). Meanwhile, adding back $Mg^{2+}$ to EGTA treated NHS did not fully restore the killing capacity of the NHS, nor was C2-depleted serum able to impact bacterial viability, suggesting that alternative pathway played a minor role in *H. pylori* killing (Fig. 2c). Finally, killing assays with a higher percent of NHS that lacked specific pathways confirmed that the classical pathway played a central role in killing *H. pylori* (Supplementary Fig. 1).

### *H. pylori* is able to resist complement in a manner that depends on lactate
Given that complement acts in the stomach to decrease *H. pylori* numbers, we hypothesized the bacterium has developed anti-complement strategies for its persistent colonization development. We speculated that *H. pylori* might sense some aspect of the gastric milieu and activate complement resistance. Although there are many candidate conditions that could be sensed, we first tested whether lactate may play a role in facilitating *H. pylori* complement resistance, since lactate has been linked to complement evasion in other

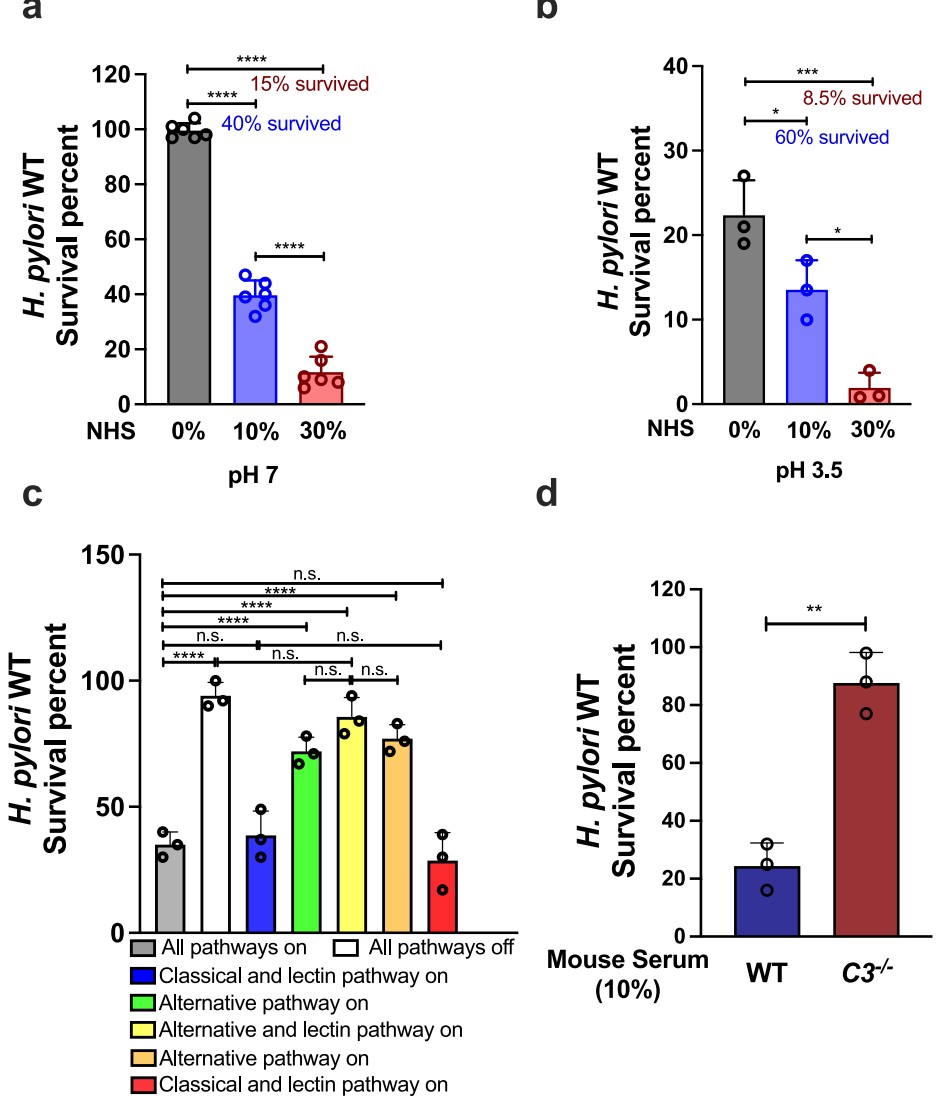

**Fig. 2 | *H. pylori* is sensitive to complement-mediated elimination.** Overnight cultures of *H. pylori* PMSS1 WT were treated with inactive and active NHS or mouse serum at the indicated final percentages for 1 h, and then viable bacterial CFUs were determined by plating. The survival percent was determined by comparing the number of CFUs from active serum divided by the total CFUs, obtained from heat-inactivated serum treatment. **a** Comparison of different final percentages of NHS at pH of 7. **b** Comparison of different final percentages of NHS at pH of 3.5. Survival ratio without NHS treatment was acquired by comparing the bacterial number at the end time point to the input number. **c** *H. pylori* cultures were treated by untreated NHS (gray bar, all pathways on); EGTA-conditioned NHS to block all complement pathways (white bar, all pathways off); EGTA + Ca²⁺ (blue bar, classical and lectin pathway on); EGTA + Mg²⁺ (green bar, alternative pathway on); C1q- depleted human serum (yellow bar, alternative and lectin pathway on); C2-depleted human serum (light orange bar, alternative pathway on only); or Factor-B-depleted human serum (light red bar, classical and lectin pathway on). All tests were carried out with 10% serum. **d** *H. pylori* cultures were treated by 10% final percentage of extracted mouse serum from C57BL/6J WT or *C3⁻ᐟ⁻* mice. In all panels, results were presented as survival percent ± standard deviation (SD), and were derived from at least triplicate biological samples with a triplicate of each given sample. The *p* values were obtained by one-way ANOVA with Tukey's multiple comparisons test (**a**, **b**, **c**), and a two-tail student *t*-test (**d**). The significance was indicated as * ($p < 0.05$), ** ($p < 0.01$), *** ($p < 0.001$), **** ($p < 0.0001$), or n.s. (not significant). Source data and exact *p* values are provided in the Source Data file.

microbes[19] and has recently emerged as important in *H. pylori* physiology[20,21,23]. We therefore grew *H. pylori* with supplementation of physiological concentrations of L-lactate[31,32], and exposed the bacteria to NHS. Strikingly, the bacterial survival rate was significantly elevated, from ~40% to ~80%, (Fig. 3a), a response that was robust even in higher level of NHS exposure (Fig. 3b). Likewise, such an effect was detected from mouse serum exposure, raising the survival from ~22% to ~80% (Fig. 3c). Lactate mainly protected *H. pylori* from classical pathway activation under different levels of serum exposure, suggesting that anti-classical pathway is the major effect (Fig. 3d; Supplementary Fig. 1). Finally, we determined that the lactate effect happened rapidly, requiring only 1 h of lactate exposure before NHS treatment to protect 70% of bacteria from elimination (Fig. 3e), a number that was similar to

long-term lactate exposure (Fig. 3a). Since lactate induction was not yet been reported during *H. pylori* gastric infection, we evaluated whether L-lactate is present in the mouse stomach and how levels are affected by infection, by isolating tissue and quantifying total L-lactate levels. In uninfected WT or *C3⁻ᐟ⁻* mice, L-lactate levels were nearly the same, at 5 μmol/gram tissue, in both corpus and antrum (Fig. 1g, h). After two weeks of *H. pylori* infection, the levels were elevated 2-3 fold in the antrum and ~1.5-fold in the corpus (Fig. 1g, h). This result supports the idea that *H. pylori* can be protected by gastric L-lactate during infection, possibly more-so in the antrum than corpus. Overall, these results suggest that lactate exposure occurs in vivo and allows *H. pylori* to rapidly adopt a state that is resistant to both human and mouse complement.

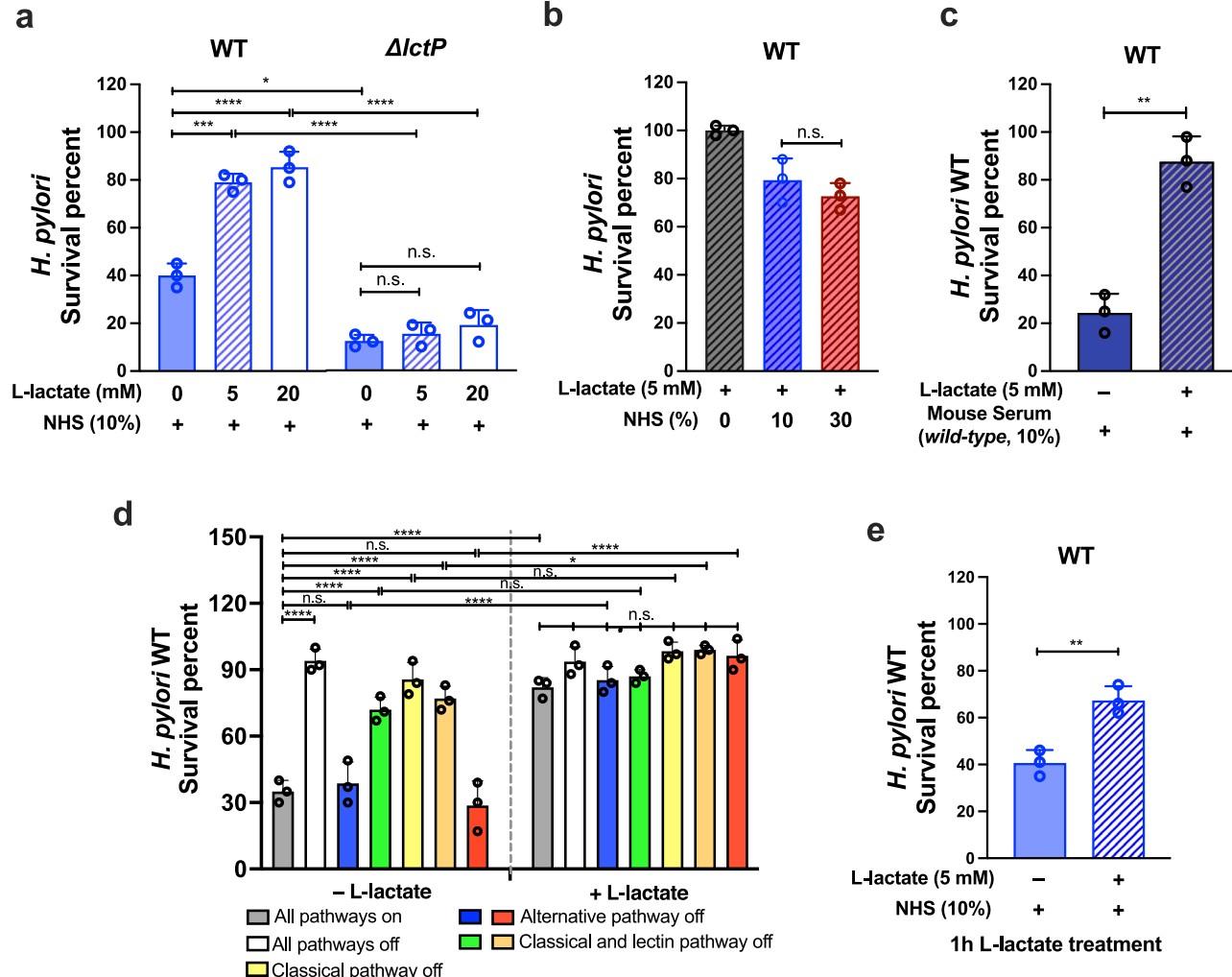

**Fig. 3 | L-lactate protects *H. pylori* from complement-mediated elimination.** Overnight cultures of *H. pylori* PMSS1 WT or Δ*lctP* mutant with or without L-lactate supplementation overnight or for 1 h, as indicated, were treated by inactive and active serum for 1 h, and then viable bacterial CFUs were determined by plating. The survival percent was determined as described in Fig. 2. **a** Overnight cultures of *H. pylori* WT or Δ*lctP* mutant grown with the indicated concentration of L-lactate were treated by inactive and active 10% NHS. **b** Overnight cultures of WT *H. pylori* with 5mM L-lactate supplementation was treated by indicated percentage of NHS. **c** WT *H. pylori* cultures were treated with C57BL/6J mouse serum. **d** *H. pylori* cultures were treated by NHS only (gray bar), EGTA-conditioned NHS (white bar), C1q-depleted human serum (yellow bar), C2-depleted human serum (orange bar),

Factor-B-depleted human serum (red bar), EGTA + Ca²⁺ (blue bar), or EGTA + Mg²⁺ (green bar) as in Fig. 2c. Survival ratio in the absence of L-lactate is identical to Fig. 2c. All tests were applied with 10% serum. **e** WT *H. pylori* cultures were treated with L-lactate for 1 h before exposed to NHS. In all panels, results were presented as survival percent ± standard deviation (SD), and were derived from triplicate biological samples with a triplicate of each given sample. The $p$ values were obtained by one-way ANOVA with Tukey's multiple comparisons test (**a**, **b**, **d**), and a two-tail student $t$-test (**c**, **e**). The significance is indicated as * ($p < 0.05$), ** ($p < 0.01$), *** ($p < 0.001$), **** ($p < 0.0001$), or n.s. (not significant). Source data and exact $p$ values are provided in the Source Data file.

## L-lactate mediated *H. pylori* complement resistance requires uptake

The above results show that lactate exposure allows *H. pylori* to become complement resistant, so we next determined whether this response requires lactate uptake to the cytoplasm. *H. pylori* has two lactate permeases called LctP1 and LctP2 (HP0140 and HP0141) that have been shown to promote lactate uptake in a distinct *H. pylori* strain[23]. The *H. pylori* lactate uptake pathway shares the basic same principles as those initially identified in *Bacillus subtilis* and *Shewanella oneidensis*[33,34]. To evaluate if complement resistance requires uptake, we created *H. pylori* PMSS1 mutants that lack the genes coding for both lactate permeases (Δ*lctP*). The mutation was confirmed by PCR and the mutants analyzed for their response to L-lactate. The Δ*lctP* mutant strain was not able to acquire a growth advantage from L-lactate, as compared to WT *H. pylori* (Supplementary Fig. 2), supporting that the LctP transporters function as

predicted in lactate uptake[23]. When treated with NHS, the Δ*lctP* strain had very low survival, of only ~8%, fivefold lower than WT (Fig. 3a). Addition of L-lactate did not improve this strain's survival (Fig. 3a). These results suggest that the anti-complement process in *H. pylori* requires L-lactate uptake.

## L-lactate prevents accumulation of C4b on the *H. pylori* surface

We next asked what step of complement activation was prevented by the lactate-treatment. Because lactate mainly protected *H. pylori* from the classical but not alternative pathway, we focused on the proteins and steps that are found only in the former (Fig. 1a). C4 cleavage is a key step of the classical/lectin pathways activation, generating two fragments: the larger C4b fragment that covalently binds to the target cell surface, and the smaller peptide C4a that is released (Fig. 4a)[35,36]. We thus tested how L-lactate affects C4 activation. *H. pylori* was co-incubated with NHS, and the amount of C4b

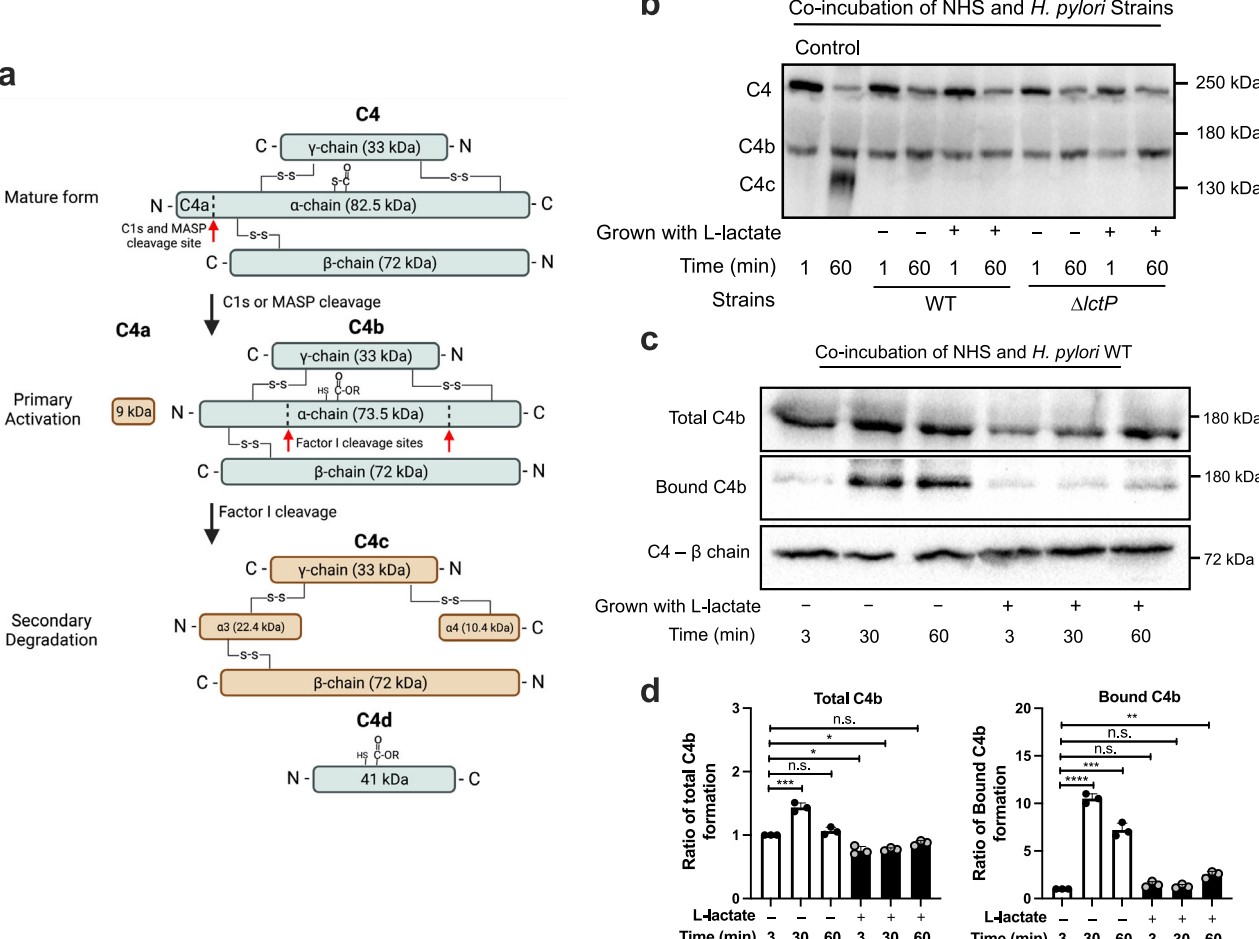

**Fig. 4 | L-lactate mediates destabilization of C4b binding to *H. pylori*.**
**a** Schematic diagram of C4 primary activation and secondary degradation, adapted from[35]. **b** Western blot of C4 protein and products after activation in the presence of WT and Δ*lctP H. pylori* over 60 min, using non-reducing gels. A positive control of complement activation and C4c formation was triggered by heated human IgG. Positions of C4, C4b, and C4c are shown at the left, and molecular weight is indicated on the right. The result is representative of three independent experiments with triplicate biological samples. **c** *H. pylori* PMSS1 WT was grown ± L-lactate and then treated by 10% NHS for the indicated time periods. Each sample was split into three: (1) treated by sample buffer only (total C4b); (2) centrifuged at 3000 × *g* for 3 min before adding sample buffer (bound C4b); (3) treated by sample buffer plus

the reducing agent beta-mercaptoethanol (BME) (C4 β-chain). All protein samples were run on 5-7% SDS PAGE and blotted with anti-C4 antibody. **d** Quantification of the experiment shown in (c), from three independent experiments. Bands were quantified by normalizing each band to the amount of total C4b or bound C4b at 3 min (leftmost band in each treatment group) using the Bio-Rad Image Lab software. Each point represents a biological sample for an independent experiment. The data were combined from the three independent experiments (represented as mean ± SD). The *p* values were obtained from unpaired two-tailed *t*-tests. The significance is indicated as * (*p* < 0.05), *** (*p* < 0.001), **** (*p* < 0.0001), or n.s. (not significant). Source data and exact *p*-values are provided in the Source Data file.

formation monitored over time by western blotting. C4 was rapidly activated in the presence of *H. pylori*, as monitored by C4b production and a decrease in total C4 at the end of the incubation period (Fig. 4b). C4b accumulation at the end of the incubation period (60 min) was not different with or without lactate in WT *H. pylori*, nor did it change between WT and Δ*lctP* strains (Fig. 4b). This outcome suggests that L-lactate does not prevent C4b formation, and therefore does not affect steps upstream of this event. We did note that L-lactate may slow down C4b formation in *H. pylori*, as evidenced by the slower accumulation of total C4b at early time points (Fig. 4c, d). After C4 cleavage, C4b deposits on the cell surface. We therefore examined the status of C4b deposition. WT *H. pylori* was treated as above, and then bacteria were pelleted to estimate bound-C4b using western blotting. In this case, there was much less stably bound-C4b associated with lactate-grown *H. pylori* as compared to non-treated bacteria (Fig. 4c). Control experiments showed that similar amounts of *H. pylori* were pelleted in all samples (Supplementary Fig. 3). L-lactate was not able to block C4b binding to the surface of the Δ*lctP*

mutant (Supplementary Fig. 4). A well-known complement resistance mechanism that acts at the C4 step is the use of the host protein Factor I with cofactors, such as C4 binding protein (C4BP)[2,37,38]. C4BP and Factor I result in cleavage of C4b to two alternative forms: the bound but inactive C4d fragment and the released C4c fragment (Fig. 4a). To determine whether this pathway was operating, we monitored C4c formation, recognized by our anti-C4 antibody (Fig. 4b). C4c was detectable in the positive control samples, but nearly undetectable in either lactate-grown or non-lactate-grown WT and Δ*lctP H. pylori* (Fig. 4b), suggesting Factor I and C4BP were not acting. Considering that the Factor I-cofactor mechanism is the only known host regulatory mechanism that acts on C4[2,37], our findings suggest that a different resistance mechanism is taking place, possibly one that is bacterial-intrinsic. Overall, these results are consistent with a model in which lactate uptake results in a bacterial response that prevents the accumulation of C4b on the bacterial surface; blocking this step prevents the formation of the MAC and subsequent cell lysis.

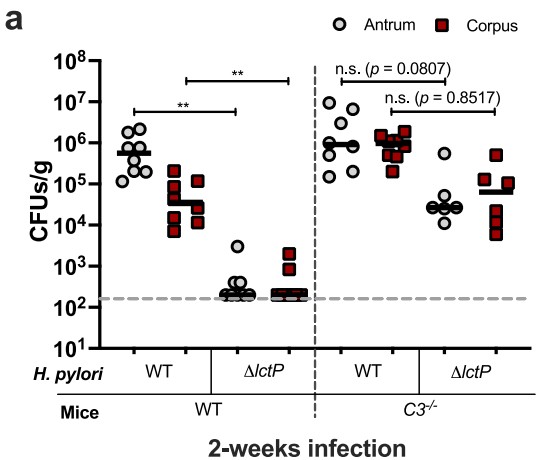
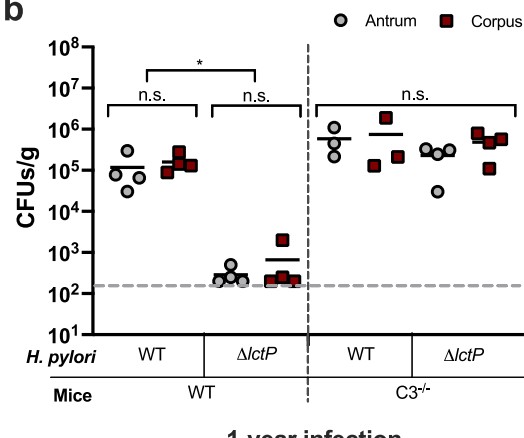

**Fig. 5 | Loss of *lctP* renders *H. pylori* colonization deficient in a manner that is recovered by loss of complement. a** Two-weeks infection by *H. pylori* strains. Group of C57BL/6J WT (*n* = 8) and isogenic *C3*⁻/⁻ (*n* = 8) mice were infected with the *H. pylori* PMSS1 WT; another group of C57BL/6J WT (*n* = 6) and isogenic *C3*⁻/⁻ (*n* = 6) mice were infected with the *ΔlctP* strain. **b** One-year infection by *H. pylori* strains. Group of C57BL/6J WT (*n* = 4) and isogenic *C3*⁻/⁻ (*n* = 3) mice were infected with the *H. pylori* PMSS1 WT; another group of C57BL/6J WT (*n* = 4) and isogenic *C3*⁻/⁻ (*n* = 4) mice were infected with the *ΔlctP* strain. The WT and *C3*⁻/⁻ mice were cohoused at weaning for at least 4-weeks before the infection and remained cohoused for the whole experimental period. At the indicated time points, the stomachs were collected and bacterial CFUs were determined by plating. Each point represents one mouse sample, with black bars representing the mean. Gray dotted line represents the limit of detection. The *p* values were obtained by Tukey's multiple comparisons test. The significance was indicated as * (*p* < 0.05), ** (*p* < 0.01), and n.s. (no significance). Source data and exact *p* values are provided in the Source Data file.

## Complement resistance is critical for stomach colonization

Our above results predict that *H. pylori* that does not mount a complement resistance response would have a severe colonization defect. We thus determined the performance of the *ΔlctP* mutant in the stomach by orally infecting WT mice. Remarkably, *ΔlctP* mutant was nearly unable to colonize the stomach (Fig. 5a). We confirmed this defective phenotype in another commonly used WT mouse model, C57BL/6N mice[26,39] (Supplementary Fig. 5). Thus, we concluded that lactate uptake promotes *H. pylori* infection.

The defect of the *ΔlctP* strains could be due to either loss of lactate-dependent growth or complement resistance. To differentiate these two possibilities, we infected *ΔlctP* mutants into *C3*⁻/⁻ mice. We discovered that the CFUs of the *ΔlctP* mutant were significantly elevated in the corpus, compared to those in WT mice, achieving levels that were not different from those of WT *H. pylori* (Fig. 5a). Because *H. pylori* creates chronic infections, we assessed how long-term loss of complement would affect *H. pylori* colonization. We allowed *H. pylori* WT and *ΔlctP* mutant infections in both WT and *C3*⁻/⁻ mice to persist for one year. In this case, we found that loss of C3 allowed sustained recovery of the *lctP* mutants to WT levels (Fig. 5b), similar to the outcome after two weeks. To minimize any gastric microbiota differences between WT and *C3*⁻/⁻ mice, we cohoused these two types of mice starting at weaning, when the microbiota is not established, and throughout the infection period. This type of approach has been demonstrated to promote the equalization of intestinal microbiota[40]. WT *H. pylori* colonization exhibited no statistically significant differences between cohoused (Fig. 5a) and non-cohoused WT or *C3*⁻/⁻ mice (Fig. 1c), suggesting no substantial influence of the resident microbiota. Taken together, these data support the idea that L-lactate uptake has a major role to protect *H. pylori* from host complement, which is critical for short-term and long-term infections.

## Discussion

We report here that the complement system plays an important role in gastric defense by mediating bacteriolytic activity. We uncovered that this detrimental effect was diminished by lactate uptake-mediated destabilization of C4b at the *H. pylori* surface. By infecting complement-deficient mice with lactate uptake deficient *H. pylori* strain, we further demonstrated that a major role of lactate utilization is to promote complement resistance (Fig. 6). Because complement plays both immune-activation and immune-regulatory functions, our work suggests the idea that pathogens have optimized particular methods of complement inactivation that protect themselves but also allow some actions of complement to proceed.

Previous work had identified that there were some complement components in the stomach, intestine, and colon, but it was not clear how these contributed to host defense. Indeed, the prevailing view was that many of these complement proteins were found in dysregulated or chronic infection situations such as Crohn's disease, ulcerative colitis, or chronic gastritis[6–8], but with an unknown function in acute infection. Complement proteins have also been detected in the intestinal tracts of healthy hosts[5]. In the case of gastric infection, previous work had found that complement depletion decreased the severity of *Helicobacter felis* induced gastritis in an IL-10 deficient mouse model[41]. However, it was not clear whether complement played a role in wild-type animals during early infection. Our work clearly shows that eliminating complement from wild-type background, by loss of C3, increases the *H. pylori* load. It was noticed that both overall glandular occupancy by *H. pylori* and the numbers per occupied gland were elevated in complement-deficient mice compared to that in WT mice (Fig. 1d, e). Our results thus suggest that complement may play an important role in glandular defense, especially in corpus region, with *H. pylori* utilizing L-lactate to thrive in these glandular regions. Future studies will be important to more precisely elucidate how complement protects gastric glands from microbial infection.

This effect of loss of complement occurred in both the antrum and corpus, but was greater in the latter. This bacterial load difference could be due to higher complement levels in the corpus, or alternatively, higher complement resistance in the antrum. Studies investigating the prevalence of subregional complement components in the stomach reported some compounds, e.g. activated C3, were more abundant in antrum and others, e.g. the membrane attack complex, were deposited equally between the two regions[8]. We noticed similar complexity from examining mouse samples. In future studies, a systematic analysis would be needed to elucidate whether there is regional variation in complement activities and how this differentially impacts local *H. pylori* viability. Interestingly, our work showed that *H. pylori* infection elevated L-lactate levels more in the antrum than the

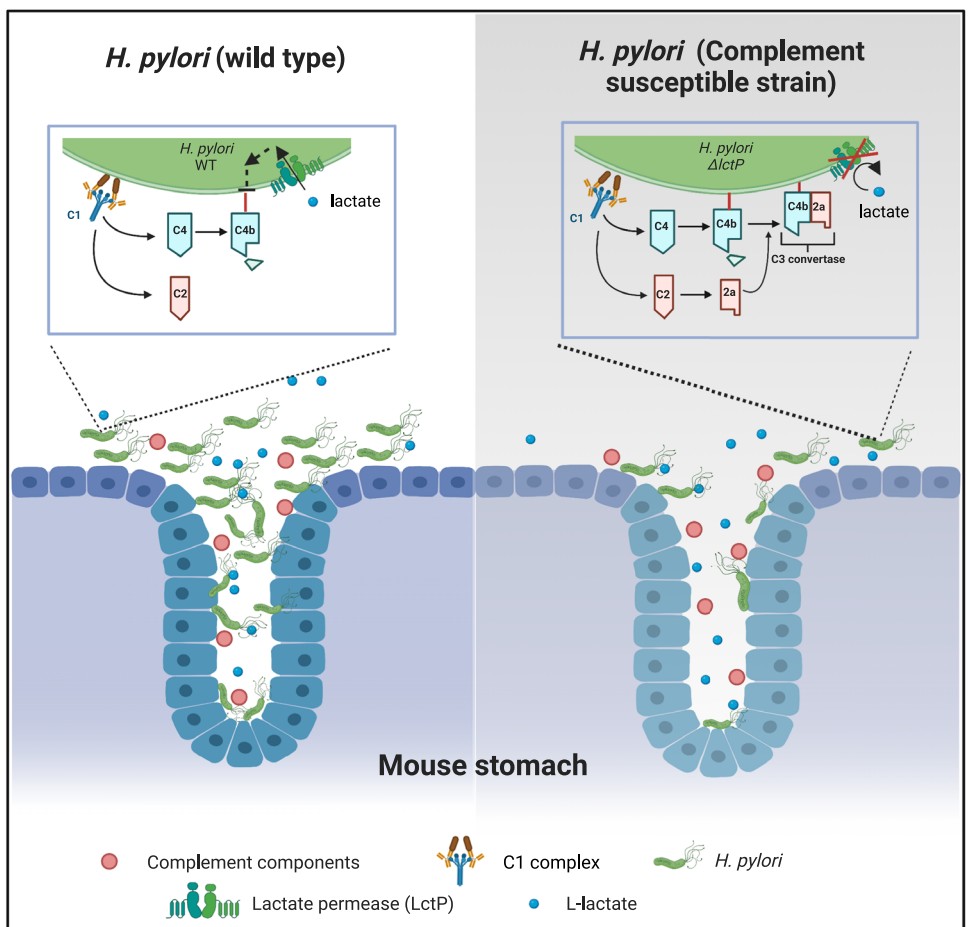

**Fig. 6 | Proposed model for L-lactate promotion of *H. pylori* gastric colonization.** Model showing how complement may operate in the stomach to inhibit *H. pylori* colonization. The left panel shows WT *H. pylori*, which is able to colonize the gastric mucosal surface and glands in the face of complement (red circles). The inset details how L-lactate (blue circles) uptake into *H. pylori* blocks the step of C4b surface deposition, rendering *H. pylori* complement resistant. This mechanism of complement resistance is distinct from that used by other bacteria, acting to further block downstream C3 convertase formation and other activities. The right gray panel shows what happen when *H. pylori* is unable to take up L-lactate. In this case, the *H. pylori* is significantly less capable of destabilizing surface bound C4b, rendering the strain susceptible to complement and less able to colonize the gastric glands and surface.

corpus. This finding suggests that *H. pylori* may be more complement resistant in the antrum due to high lactate levels, and thus does not show a great effect of C3 loss. Another key aspect of the corpus is its lower pH compared to the antrum. Our results show that complement retains the ability to kill *H. pylori* killing even at low pH. Low pH itself is harmful to *H. pylori*, so this outcome may suggest that although complement has reduced activity at low pH[42], there is still enough activity to decrease *H. pylori* numbers.

Furthermore, *H. pylori* resides and multiplies in the local niches where the pH is closer to neutral[43]. Although the human stomach is more acidic than the mouse one, *H. pylori* resides at the near-neutral epithelial surface, and thus the species-specific pH differences may not have a great effect on complement functions in *H. pylori* colonizing niches and suggests that our findings would apply well to humans. Complement lowers bacterial numbers by several mechanisms, including direct lysis and opsonization. We propose that cell lysis dominates the complement-mediated activities in vivo because there are few immune cells present in the stomach during *H. pylori* early infection[28]. Indeed, immune cells are only recruited above basal levels after ~3 weeks of infection[28]. There may be subtle effects that we cannot rule out or detect, however, due to low level immune cell recruitment and effects of C3 loss on anaphylatoxin release (C3a and C5a)[2–4]. Our finding that *H. pylori* blocks C4b deposition supports this proposal, because a blockade at this step would also prevent creation of the C3a and C5a products, which are immune cell chemoattractants.

Our work shows that *H. pylori* has a previously undescribed way of blocking complement. We found lactate-exposed *H. pylori* had much reduced bacterial-associated C4b, which is not due to Factor I cleavage. Whether this strategy is widespread remains to be seen, as there have been very few studies on the interplay between microbes and complement in the GI tract. Whether there are connections to lactate in other microbes is also a unexplored area, although several recent studies have identified lactate as a critical nutrient to promote bacterial expansion in GI tract[44]. Additional studies will be needed to determine the mechanism by which *H. pylori* decreases C4b deposition. C4b has a labile internal thioester bond that reacts broadly with amino and hydroxyl groups[36], so it seems unlikely that lactate-treatment removes the target of C4b. Instead, it seems more likely that *H. pylori* removes the C4b, possibly by rapidly shedding the target molecules or employing some type of protease (Fig. 6). This complement resistance mechanism is in addition to several other previously described *H. pylori* ones that utilize host proteins or carbohydrates. For example, *H. pylori* employs host CD59 to block MAC formation[14]. The *H. pylori* virulence factor SabA binds sialic acid, which has high affinity to Factor H. Binding of Factor H allows *H. pylori* to activate Factor I for C3b cleavage, a process to block alternative pathway activation[45,46]. It indeed seems that *H. pylori* has complex and robust

complement resistance mechanisms to cope with this host defense system, emphasizing the importance of complement in *H. pylori* biology.

Another finding reported in this work is the expansion of our knowledge of how *H. pylori* utilizes lactate. Lactate utilization supports microbial infection in various ways, including both as a growth substrate and for other reasons. For example, *Salmonella Typhimurium* utilizes L-lactate as a nutrient during gut infection[44]. *Neisseria*, another example, utilizes lactate to promote complement resistance, as we show for *H. pylori*[19,47]. However, key differences exist between these two bacterial systems. *N. meningitidis* utilizes lactate to evade alternative pathway activation by stimulating the sialic acid biosynthesis pathway to robustly recruit Factor H, which facilitates Factor I to cleave C3b[8,19]. In *H. pylori*, L-lactate did not promote *H. pylori* LPS sialylation from our preliminary investigation, but exclusively mediated classical pathway defense by promoting the removal of C4b. One atypical aspect of *H. pylori* is that it has duplicated *lctP* genes (*lctP1* and *lctP2*) on the *H. pylori* genome[19,23]. Overall, these findings indicate a divergent bacterial environmental adaptation from a common nutrient acquisition origin.

The mechanism that connects lactate to the anti-complement response in *H. pylori* is not yet clear. Results presented here show that *H. pylori* promotes lactate production in the stomach. The source of lactate is not yet known, but many fast growing cells, including epithelial cells, glandular stem cells, and activated phagocytic cells, are able to produce lactate and potentially impact local lactate concentration[48–53]. One idea is that *H. pylori* survives and colonizes the local niches where lactate is relatively abundant. An important observation that supports this idea is that *H. pylori* colonizes the progenitor and stem cells compartments and stimulates Lgr5+ stem cell proliferation[54]. Stem cells feature a high glycolytic rate compared to more differentiated cells[50], and likely produce more lactate.

More than half of the world's population is infected with *H. pylori*, which causes a series of gastric diseases[55]. Treatment relies on "triple-therapy", consisting of two antibiotics and a proton pump inhibitor[56,57]. There are increasing reports about antibiotic resistance, such that *H. pylori* has been placed on the WHO's global priority list of antibiotic resistant organisms[58,59]. Our study strongly indicates that L-lactate utilization is tightly intertwined with *H. pylori* colonization, beyond use as a carbon source[23]. Thus, we envision therapeutics that might block lactate uptake, and in turn result in robust complement-mediated elimination of *H. pylori*.

## Methods

### Ethics statement
All animal work was approved by the University of California, Santa Cruz Institutional Animal Care and Use Committee by following Protocol OTTEK1804 and OTTEK2014.

### Bacterial strains and growth condition
*H. pylori* PMSS1[25] and isogenic mutant were used for these studies (Supplementary Table 1). The isogenic *ΔlctP* mutant (KO1710) was constructed using a PCR-based method as described previously[60]. To delete the lactate permease coding genes, the coding region of both *hp0140* and *hp0141* were entirely replaced by a chloramphenicol resistance cassette (*cat*), selected by chloramphenicol (25 μg/ml) and the mutants verified by PCR amplification. The primer set is provided in the Supplementary Table 2. For mouse model infections, GFP-expressing PMSS1 WT and *ΔlctP* strains were constructed by natural transformation with plasmid pTM115[26] to create PMSS1 pTM115 (KO1568) and PMSS1 *ΔlctP* pTM115 (KO1711). pTM115 has previously been shown to be stably maintained during infection with robust high level GFP expression[26]. For the transformation, plasmid isolated from *H. pylori* strain SS1 pTM115[26] was used with selection CHBA plates containing 15 μg/ml kanamycin.

For solid media, *H. pylori* strains were grown on Colombia Horse Blood Agar containing: 5% defibrinated horse blood (Hemostat Laboratories, Dixon, CA), 0.2% β-cyclodextrin, 10 μg/ml vancomycin, 5 μg/ml cefsulodin, 2.5 U/ml polymyxin B, 5 μg/ml trimethoprim, and 8 μg/ml amphotericin B (CHBA) (all chemicals are from Thermo Fisher or Gold Biotech). For antibiotic resistance marker selection, bacterial media was additionally supplemented with 25 μg/ml chloramphenicol or 15 μg/ml kanamycin. For liquid media, *H. pylori* cultures were grown in brucella broth (BB) medium supplemented with 10% heat-inactivated fetal bovine serum (FBS) (BB10) (Life Technologies). *H. pylori* were grown under microaerobic conditions of 10% $CO_2$, 5% $O_2$ and 85% $N_2$ at 37 °C.

### *H. pylori* growth assays
CHBA-maintained *H. pylori* strains were resuspended in BB10 media at an initial optical density 600 nm ($OD_{600}$) of 0.1. Sodium L-lactate (Sigma) was added to a final concentration of 10 mM. 100 μl of this bacterial culture were placed in wells of 96-well plates and grown with 200 RPM constant shaking, at 37 °C, 5% $O_2$ and 10% $CO_2$ atmosphere, using a FLUOstar microplate reader (BMG Labtech). $OD_{600}$ was measured using the microplate reader during growth.

### Serum resistance assays
Normal human serum (NHS, Sigma), negative for anti-*H. pylori* antibodies as indicated by manufacturer, was used for *H. pylori* serum sensitivity tests. The NHS was either left untreated (active) or heat-inactivated by heating in a water bath for 60 min at 56 °C. For serum-mediated killing assays, overnight bacterial cultures were diluted to $OD_{600}$ of 0.1 (~$10^7$ bacteria) in fresh matched medium (BB with or without lactate), and then NHS (active or inactive) was added to the final volume percentage of 10% or 30%. In some cases, complement pathways were queried by addition of 10 mmol/L EGTA (Sigma), 12 mmol/L $CaCl_2$ or 12 mmol/L $MgCl_2$ to NHS (all final concentrations)[12] or by using C1q-, C2-, and Factor-B-depleted human serum (Complement Technologies). Samples were incubated for 1 h under microaerobic conditions with shaking. After the desired time, samples were serially diluted and plated on CHBA, and incubated for five days to allow colony formation. The bacterial survival ratio was calculated by determining the number of colony forming units (CFUs) after exposure to either active NHS or to inactive NHS, and then dividing the numbers in the active serum by those in the inactive.

### Mouse infection
Male and female C57BL/6J mice (Helicobacter-free, The Jackson Laboratory) and B6;129S4-C3<tm1Crr>/J (*C3*$^{-/-}$) mice (Helicobacter-free, The Jackson Laboratory), or female C57BL/6 N mice (Helicobacter-free, Charles River Laboratories), were housed at the animal facility of the University of California Santa Cruz.

Mice were between 7 and 9 weeks old at the time of *H. pylori* infection. For some infections, *C3*$^{-/-}$ mice were cohoused with their wild-type parental C57BL/6 J mice at weaning for at least 4-weeks before the infection and remained cohoused for the whole experimental period. All animals were orally infected via a 20-gauge 1.5-in. feeding needle (Popper) with 100 μl containing ~$10^8$ GFP+ *H. pylori* strains. These strains were prepared by liquid growth to late-exponential phase in BB10 medium. After the infection period, the animals were sacrificed via $CO_2$ narcosis, the stomach dissected, opened along the lesser curvature, and the forestomach region removed and discarded. The remaining tissue was divided into three parts—the corpus, the transition zone, and the antrum, using difference in tissue coloration as a marker[61]. The transition zone was discarded and not included in experimental analysis. Each region was then divided into two pieces with 0.3–0.5 cm$^2$ of each, one piece for total bacterial number counting and lactate concentration measurement, and the other for gland isolation.

### Total bacterial number counting in mouse stomach

Mouse gastric tissue was weighed, homogenized using the Bullet Blender (Next Advance) with 1.0-mm zirconium silicate beads, diluted and plated to determine the number of CFUs/gram of stomach tissue on CHBA with the addition of 20 μg/ml bacitracin, 10 μg/ml nalidixic acid, and 15 μg/ml kanamycin.

### Lactate concentration measurement

The homogenized tissue lysates were then performed by using Lactate Colorimetry Assay Kit (Biovision, Milpitas, CA) according to manufacturer's instruction.

### Gland isolation

Dissected gastric tissue was cut into 1-mm$^2$ pieces and incubated in Dulbecco's phosphate-buffered saline (DPBS) (Millipore) plus 5 mM EDTA at 4 °C for 1 h. The tissues were then transferred into ice-cold DPBS containing 1% sucrose and 1.5% sorbitol and shaken roughly by hand for 1–2 min. The isolated glands were allowed to settle, and were kept on ice until examined.

### Gland analysis and microscopy

Isolated glands were labeled with 10 μg/ml Hoechst DNA stain (Life Technologies). Ten microliters of labeled glands were placed on glass slides and visualized using a Nikon Eclipse E600 microscope with fluorescence filters for 4′,6′-diamidino-2-phenylindole (DAPI) and GFP. For each time point of infection, 100 glands were imaged for the corpus and 100 for the antrum, and the number of intra-gland *H. pylori* was manually counted for each gland. Gland occupancy was calculated as the percentage of glands occupied per mouse. Gland load were calculated by averaging the number of bacteria in occupied glands per mouse.

### Mouse gastric pH measurement

The pH measurement was performed immediately after stomach dissection. The forestomach region was removed to allow the electrode of pH meter to enter the interior regions of stomach and closely touch the inner wall of either corpus or antrum subregion. The pH value was determined using pre-calibrated portable pH meter LAQUA with a 9618S-10D Micro ToupH electrode (3 mm diameter) (HORIBA, Ltd, Kyoto, Japan).

### Mouse blood collection

Housed *Helicobacter*-free C57BL/6J mice were used for blood serum extraction. CO$_2$ narcosis was performed to euthanize the mice. Cardiac puncture was carried out using a 3 cc syringe with a 22 gauge × 1″ needle. The needle was advanced in the notch to the left of the mouse's xiphoid, into the chest, and heart. Once the needle was in the heart, the plunger was withdrawn to syringe generate slight back pressure to allow blood flow into the syringe. Each mouse enabled ~800 cc blood collection. The collected blood was allowed to clot for 1 h on ice without any coagulant inhibitors. The clotted material was removed by centrifugation at 3000 rpm for 15 min. The top serum was collected and stored at −80 °C for future analysis.

### Serum C4 activation assay

Overnight *H. pylori* PMSS1 WT and mutants were suspended into BB media containing ~10$^7$ bacteria. Active NHS was added to the bacterial culture and allowed to co-incubate for indicated time periods at 37 °C. As positive control for complement classical pathway activation and the creation of C4c, human IgG (Invitrogen) was heated at 63 °C for 20 min to trigger aggregation. One milligram per milliliter heated IgG (final concentration) was co-incubated with NHS to activate complement[12]. Reactions were stopped by adding 6x sample buffer (0.3 M Tris-HCl, pH 6.8, 6 M glycerol, 10% w/v SDS, 9 mM bromophenol blue). Some protein samples were further treated with final 5%

β-mercaptoethanol to cleave disulfide bonds. All samples were then incubated at 90 °C for 10 min. C4 activation was analyzed by monitoring the formation of cleaved C4 portion, C4b, through western blot.

### Western blot

Samples were separated on 5% or 7% (wt/vol) acrylamide SDS/PAGE gels, and then used for immunoblotting. For immunoblotting, gels were soaked in transfer buffer [48 mM Tris-base, 39 mM glycine, 1.3 mM SDS, 20% (vol/vol) methanol] for 5 min and then transferred to an immunoblot polyvinylidene fluoride (PDVF) membrane (Bio-Rad) for 60 min at 25 V through Trans-Blot® SD Semi-Dry Transfer Cell (Bio-Rad). The membrane was blocked for 1 h with blocking buffer (PBS with 5% milk plus 0.2% Tween-20) at room temperature. Primary antibody was added and incubated for 16–18 h at 4 °C using the 1:1000 dilution for Rabbit anti-human C4 antibody (mAb #60059) (Cell Signaling). After incubation, the membranes were washed and HRP-conjugated Goat anti-rabbit added at 1:10,000. After incubation, the membranes were washed and treated with HRP substrates (SuperSignal West, Pico and hydrogen peroxide mixed at 1:1 ratio; Thermo Scientific). Blots were then visualized using a Chemidoc imaging system (Bio-Rad). Uncropped blots are provided in the Source Data file.

### Statistical analysis and reproducibility

All statistical tests were performed using GraphPad Prism version 9 as indicated in the figure legends. Differences in *p* value of <0.05 were considered significant. No statistical tests were used to determine sample size. For in vitro, and in vivo experiments, samples sizes were determined based on the numbers required to achieve statistical significance using indicated statistics, with a minimum of three independent experimental performance to ensure data reproducibility. All in vitro, experimental treatment group allocation was random. Allocation of mice was random in all in vivo experiments, taken from littermates. All studies were performed in University of California, Santa Cruz.

### Cartoon schematics

The cartoon schematics (Figs. 1b and 6) were created with BioRender.com.

### Reporting summary

Further information on research design is available in the Nature Portfolio Reporting Summary linked to this article.

## Data availability

The authors declare that the data supporting the findings of this study are available within the paper and its Supplementary Information files. Source data are provided with this paper.

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

## Acknowledgements

The authors are grateful to members of the Ottemann Lab for multiple discussions, and Isabel Arnold (University of Zurich), and Vicki Auerbuch Stone (University of California Santa Cruz) for their thoughtful comments on the manuscript. The described project was supported by the National Institute of Allergy and Infectious Disease (NIAID) grants R21AI144740 and 1R01AI116946-06, and Cancer Research Coordinating Committee of California CRR-19-582281 to K.M.O. The funders had no role in study design, data collection, and interpretation, or decision to submit the work for publication.

## Author contributions

Conceptualization, methodology, investigation, writing: S.H. and K.M.O. Funding acquisition: K.M.O.

## Competing interests

The authors declare no competing interests.
