## [Peer Review File · Nature Communications]

Helicobacter pylori initiates successful gastric colonization by utilizing L-lactate to promote complement resistanceREVIEWER COMMENTS

Reviewer #1 (Remarks to the Author):

In this manuscript, Hu et al. present data that provide important additions to the literature on the complement system and *H. pylori*. The manuscript is well written and presented in a logical manner with well thought out schematics. Using a combination of mouse models, genetic manipulation of *H. pylori*, as well as altering culture conditions, the authors show that complement attenuates gastric colonization by *H. pylori* through the classical/lectin pathway. *H. pylori* can utilize host L-lactate to resist complement by blocking the accumulation of C4b, a cleavage product of C4, on the *H. pylori* surface. The following are suggestions to improve this study.

While this manuscript presents interesting and important results, with experiments that are well conducted, and data analysis that is well performed, it is lacking in supporting data of the human relevance. Could the authors look at complement in human sera from patients with and without *H. pylori* infection?

The authors looked at a 2-week *H. pylori*-infection and determined that complement restricts colonization of *H. pylori*. Does this phenotype in colonization persist at later time points of infection?

The authors show bacterial load in the C3^{-/-} mice corpus is 100-times that of the load in wild-type mice. Is the pH in the gastric corpus of both mice equivalent?

How was inflammation affected in the gastric antrum and corpus of WT mice and C3^{-/-} mice with and without *H. pylori* infection? It has been shown that lower levels of bacterial colonization can be correlated with increased gastric inflammation.

Reviewer #2 (Remarks to the Author):

Review for Hu and Ottemann, 2022, Nature communications

Using a combination of *in vivo* and *in vitro* methods, Hu and Ottemann present an exciting discovery that *H. pylori* colonisation and complement resistance is significantly affected by L-lactate uptake and interestingly, that complement seems to particularly affect colonisation in the corpus region of the stomach, compared with the antrum. The data supports a novel mechanism of complement resistance in which lactate uptake decreases C4b deposition on the bacterial cell surface. The mechanism for how lactate enables *H. pylori* to modify levels of C4b on the cell surface remains to be fully characterised and the authors speculate on potential mechanisms in the discussion. Overall, the data fully supports the authors conclusions, but clarification regarding Figure 4 should be included (see comments below). The authors show that this mechanism is not triggered by the alternative pathway, yet the authors do not investigate further whether lactate could be affecting the classical or lectin pathway. A suggestion to investigate this is suggested below, although it is recognised that fully characterising this may be outside the scope of this study. In summary, this paper represents a shift in our understanding of the role of complement in *H. pylori* colonisation and complement resistance. All methodology used is sound and reproducible. A limitation of this study is that all data is collected in a mouse model of infection, rather than any investigations in human samples. Have the authors considered quantifying the lactate using human gastric corpus and antrum tissue as an initial rationale for these findings being relevant to the human system? I hope that the comments below will be useful in improving the manuscript.

Line 57, Typo, worldwide to worldwise

Line 80, italicise '*in vivo*' if this adheres to journal formatting.

Line 90, C57Bl/6 conventional naming, inconsistent- Figure 1, panel B uses C57BL/6J and line 598 uses C57Bl/6 please use standard format and ensure consistency throughout.

Line 97, Gland counting – A supplementary figure showing a representative microscopy image of gland occupancy versus gland load is very important here for the reader to understand the analysis and the clarity of images you are using to establish quantification of these parameters. Have you got data to support that the plasmid expressing GFP is stable and maintained during infection? If so, this would be important to mention here.

Line 104/105- during 'the' infection period used here. Could you define this in brackets ie (14 days)?

Figure 1- intriguing observation re corpus and antrum- have the authors attempted to quantify the levels of C3 (and/or alternative complement antibody components) in a WT infected stomach perhaps by flow cytometry to understand if the quantities differ between regions, which could complement the lactate quantification findings to explain the observed phenotype?

Figure 1 title- plays 'a' vital role instead?

Line 607- error bars (E, F). There is no panel F in Figure 1- I think this is meant to say (D, E)

Figure 2, what percentage of NHS is used in Panel B? Assumption is that this is 10% but this should be clearly defined in the legend or figure itself. Also, the line surrounding the panels should be removed.

Line 619- add 'respectively' after the word activation.

Figure 3C- it is not clear to me why the authors refer the reader to panel Figure 2C to visualise the survival in 10% WT mouse serum (22%) and compare this to survival shown in Figure 3C (80%). It would be much more logical for the data of survival in 10% WT mouse serum to be a bar in Figure 3C (ie 10% mouse serum, minus L-lactate), so that a direct comparison can be visualised, without switching between figures.

Figure 3G- it would be logical to add a WT column to this graph such that survival in the WT strain can be directly compared statistically to survival in the Δ lct strain in NHS.

Figure Supp 2. Perhaps the x-axis could be labelled more clearly to indicate more time points during the 21 hour period.

Figure 4B, labelling typo, 'H. pylori stains'

Figure 4B and 4C. The results comparing total C4b accumulation seems to be telling different stories between panel 4B and 4C. When comparing C4b accumulation in panel B, there seems to be little difference between +/- lactate after 1 minute incubation and the authors therefore conclude, lactate has no impact on the accumulation of C4b. However, when looking at Panel C, +/- lactate at 3 and 30 minutes, there seems to be much less C4b in the presence of lactate versus the absence. Moreover, the quantification data in the bottom of Panel 4C shows a significant difference between Total C4b +/- lactate at 3 and 30 minutes, so the authors should clarify their statement on lines 184-186, which reads as the following..... 'C4b accumulation, however, was not different with or without lactate in WT H. pylori, nor did it change between WT and Δ lctP strains (Fig. 4B). This outcome suggests that lactate does not prevent C4b formation, and therefore 186 does not affect steps upstream of this event.' Given that this data forms part of an important conclusion, I would emphasise that this conclusion should be modified given the differences in these time points.

Figure 4C – the bottom panel on this, ie the quantification data should be labelled as Figure 4D, which will make it easier to reference to in the text.

Figure 4 legend- the conditions 2-3 for Panel C are labelled in a different order to how they appear in the gel. Please correct in the figure legend.

Typo, Figure 4 legend, 'quantification purpose' should read 'purposes'

Figure 4 – the y axis on the Panel C graph states 'Ratio of ...'. How has this ratio been calculated? In relation to what? Can this be clarified in the legend.

Line 196-198 states that C4c formation was undetectable. I would suggest that a positive control band of C4c detection is required here (even if this is a supplementary figure) in order to justify that if C4c were present, the antibody would detect this and therefore fully support the conclusion that Factor I and C4Bp are not acting in this pathway.

Figure 5. All comparisons of significance seem to be made aside from the colonisation densities in the antrum of the Δ lct strain in the WT mouse compared to the C3 knockout mouse. This should be compared for completeness.

Lines 219- the statement that co-housing enables mixing of the microbiota, can a study be referenced to support this suggestion that 4 weeks leads to changes in microbiota of co-housed mice. I am assuming this data may come from the lower gut rather than the stomach? This should be clarified given that no data is shown on the microbiota compositions between these mice.

Line 236, Typo 'tracks'

Line 245, Typo 'elevated a L-lactate'

Line 254, the statement that the C4b deposition resistance mechanism has not been reported for bacteria- can this be expanded to include information on whether the mechanism of lactate-mediated complement resistance in other bacteria (apart from *Neisseria* mentioned in the introduction) has been defined or not, or a comment as to how widespread this mechanism could be across bacteria?

Line 273, Typo 'an colleagues'

Line 355, can justification be given as to how infection-free status was confirmed in individuals from which the NHS was isolated.

Although the authors correctly conclude that C4b deposition significantly decreases by lactate, is there a possibility that *H. pylori* could be regulating the activity of MASPs in the lectin pathway and therefore the cleavage of C4 and binding of C4b to the cell surface? Can this be explored by differentiating between whether this mechanism is triggered by the antibody or lectin pathways? Perhaps the NHS could be depleted of antibody and determine sensitivities to complement +/- Lactate?

Reviewer #3 (Remarks to the Author):

The manuscript "Helicobacter pylori initiates successful gastric colonization by utilizing L-lactate to promote complement resistance" by Shuai Hu and Karen M. Ottemann describes that *H. pylori* is sensitive to complements, and in particular through the Ca²⁺-dependent classical pathway. *H. pylori* infect complement depleted mice in higher infectious loads. Uptake of L-lactate makes *H. pylori* more complement resistant by blocking deposition of the complement C4b on *H. pylori*. This is a story similar to the *Neisseria* bacteria that activates biosynthesis pathways by uptake of L-lactate for complement resistance. The authors have previously published that L-lactate is taken up by *H. pylori* in cultures and this promotes increased growth rate. Thus, it is likely that L-lactate regulates many biosynthesis pathways in *H. pylori* including the large family of membrane proteins and most likely also its LPS-composition.

My main comment is about the validity of the complement system and resistance at lower pH, ie the acidic condition in the stomach which is very different from the buffered pH in circulation where most of the complement system activation takes place, something which I hope the authors can test and sort out.

Main Comments:

1) In line 69 ,the authors mention "in Neisseria meningitidis by activating a sialic acid biosynthesis pathway that protects against complement's action¹⁸." Have the authors considered to explore this mechanism for better understanding of the complement resistance and the protection against C4b accumulation on the H. pylori bacterial surfaces?

2) Line 83, The authors state" Overall, our results reveal that complement plays an important role in controlling bacterial colonization in the GI tract" (and in the first sentence in the Discussion). But I do not think a result from the mouse stomach can be applied as a conceptual result for the entire GI tract, but suggest the authors to be a bit more modest and refer to the gastric tissue/location.

3) In line 91-92 the authors describe the length of the mice infection; "After two weeks, total bacterial counts and gland colonization were evaluated in each of the two main stomach regions, antrum and corpus". Considering that H. pylori infection are chronic and life-long in humans, it would be reasonable to infect the mice for a period that corresponds to a chronic infection, i.e., for a few months instead of an acute infection as performed here. A chronic infection might reveal conditions that would better reflect the human infection scenario.

4) In lines 94-95 the authors find that "Total bacterial load in the C3^{-/-} mice corpus was significantly elevated 100-fold compared to WT mice (Fig. 1C)." This suggest that the complements in the mouse stomach are more prevalent in the corpus region. The authors should probe the antrum vs corpus mucosa to test if there is a difference in complement levels and hence availability

5) From line 124 the authors test the complement activation by Ca for classical pathway and Mg to activate the alternative pathway and killing of H. pylori. And Hp is sensitive for killing by complement activation by Ca for classical pathway. This in vitro test looks convincing, but is performed in pH condition that are most compatible with human and mouse sera, i.e., in the environments where the complement system is supposedly active. It is well established that lower pH inactivates the complement system, see e.g. this reference from 2016 by Dantas E., et al; Low pH impairs complement-dependent cytotoxicity against IgG-coated target cells. *Oncotarget*. 2016; 7: 74203-74216.

<https://www.oncotarget.com/article/12412/text/>. And thus reduction in complement activation takes place already at pH 5-6. Thus, the authors need show that the complement activated killing of H. pylori can take place also at lower pH. The mouse experiments indeed show that H. pylori manage accomplish higher infection loads (higher CFU) in the complement depleted mice. But similar to the argument about reduced pH in the human stomach, vs test by human sera at neutral pH, it is important to remember that mice do not demonstrate low pH in the stomach, but are much higher in stomach pH compared to the human stomach. Complement activation at reduced pH in the mouse could be tested by use of the INS-GAS mice (the Jim Fox model) where human gastrin drives the acid secretion and the mice demonstrate a more acidic and hence human like stomach pH. Thus, the in vitro tests and in addition the mouse tests would benefit from testing during conditions that mimic the acidity of the human acidic stomach mucosa i.e., where the H. pylori infection is located and less in contrast to the pH of the sera/blood in circulation.

6) In line 144; the authors conclude that "Lactate protected H. pylori from the classical/lectin pathway, activated by Ca²⁺, significantly raising the survival percent from ~40% to ~85%, but not the alternative pathway, activated by Mg²⁺, which did not differ ± lactate (Fig. 3E)." and this conclusion was also reiterated in the section entitled "L-lactate prevents accumulation of C4b on the H. pylori surface." And I agree to that with no lactate added the classical pathway seems to be the active pathway in killing H. pylori. But when lactate is added the Survival-percent increases from 40 to 80 % and with so high levels, it is not possible to see any effect of the EGTA +/- Ca or Mg treatment. And since the killing mechanism cannot be interpreted from these bars, I do not think the authors can conclude that the classical pathway is active also during the conditions of lactate supplementation. Since there is less killing with lactate, the authors can rerun this experiment with higher levels of human sera to kill of more of H. pylori and hence reduce the Survival percent. A lower bar (survival) opens up for interpretation by use of the EGTA, i.e., if Ca is still necessary for complement killing of the H. pylori or if the lactate has induced another

mechanism for resistance and survival.

7) Same page, line 152, the authors deliver an interesting result "In uninfected WT or C3^{-/-} mice, L-lactate levels were nearly the same, at 5 $\mu\text{mol/gram}$ tissue, in both corpus and antrum (Supplemental Fig. 1). After two weeks of *H. pylori* infection, the levels were elevated 2-3-fold in the antrum and ~ 1.5 fold in the corpus (Supplemental Figure 1)." Thus, infection activates lactate production in the gastric tissue. I have not followed the papers on *H. pylori*/ lactate but could not find that such induction of lactate had been described previously. Please explain how come you present this result in the Supplements and not in the main Figures. Please also present the results in the context of what has been previously described about infection-activated lactate production or if this is a novel result? A better context would help me and other readers to better understand the novelty of the presented results.

8) The induced lactate in the infected tissue would also have been very interesting to study also during chronic infection, such as a few months of infection instead of the short acute infection model used by the authors.

9) Page 9, line 195, "To determine whether this pathway was operating, we monitored C4c formation, recognized by our anti-C4 antibody (Fig. 4B). C4c was nearly undetectable in either lactate-grown or non-lactate-grown WT and ΔctP *H. pylori* (Fig. 4B), suggesting Factor I and C4BP were not acting." This is of course an interesting result, but since the experimental systems are difficult, this result would warrant a positive control, i.e., the authors need show that the mechanism is indeed operational with the serum samples used for a bacteria that is well known to bind C4BP.

10) In the final section "Complement resistance is critical for stomach colonization The authors describe the C57BL/6N (6N) mice whereas in the Fig 1 the authors use C57Bl6/J and in Suppl Fig 5 the mice are described as C57BL/6N mice and as C57Bl6/N mice. Would be good to see an explanation why the authors use different breeds of C57BL/6 (in addition, the nomenclature is C57BL/6N or C57BL/6J, not C57Bl6/N or C57Bl6/J).

11) In the Discussion, line 261, "*H. pylori* virulence factor SabA is able to bind sialic acid, which facilitates blocking alternative pathway activation^{41,42}. I find this sentence /statement difficult to understand, which blocking? and which complement? pathway activation?", please explain how come you find SabA of importance in this perspective.

12) In the Discussion line 273, "Guillemin an colleagues found that the complement component C4 coding gene C4B, is upregulated upon *H. pylori* infection of gastric epithelial cells)⁴⁷. I did not find this mentioned in Guillemin's article, but maybe it is listed in a supplementary file? In that case the authors need point out where the reader can find this information.

Minor comments:

1) Page 3, line 51-52 describes the C3 to C3B whereas in Fig 1A. the C2 to C2B are illustrated. This goes also for Fig 6 which illustrates C2 but no C3. This needs some better coordination/ description to fit the text .

2) Line 57; "worldwise"

3) Line 64 the authors mention "complement evasion mechanisms unlike found in other microbial species¹⁴". I think it would be appropriate to describe these "other mechanism" to some extent either in the Introduction or in the Discussion. The Discussion is rather vivid and voluminous but does not explain complement resistance mechanisms in other well explored pathogens.

4) From line 124 the authors test the complement activation by Ca for classical pathway and Mg to activate the alternative pathway. But this is somewhat confusing since in the Figure 2 "*H. pylori* is sensitive to complement mediated elimination", the similar text can easily be misunderstood since Mg and Ca comes in the wrong order: "(B) Mg²⁺ or Ca²⁺ were individually added back to the culture to resume either the classical/lectin or alternative pathway activation".

Reviewer #4 (Remarks to the Author):

General comments

The authors have developed a model that describes how *Helicobacter pylori* becomes resistant to complement (C) and therefore thrives in the stomach of experimental mice.

They show that *H. pylori* multiply in the corpus of the stomach when complement activation is truncated at the level of C3 (---tantamount to C resistance)

They also propose that, *in vitro*, *H. pylori* are susceptible to killing by C in normal human serum (NHS), exclusively by the classical pathway of C. Killing is reversed (nullified) by the presence of lactate because of resultant disruption of the covalent binding of C4b to the surface of the bacteria, thereby halting progression of C activation through to insertion of MAC (C5b-9) that would result in killing of the organism. The evidence given for C4b disruption is solid!

The experiments are carefully performed and the use of the Δ lctP mutant ensures that effects of lactate are specific for their observed phenotypic effects on survival ("serum survival phenotype") of WT *H. pylori* in serum. Lacking direct confirmation that *H. pylori* survive *in vivo* (in the mouse) and are resistant to serum killing *in vivo* because of lactate uptake, inoculation of mice with the Δ lctP mutant to demonstrate reduced or nonviability of this strain *in vivo* serves as a proxy (control) for *in vivo* specificity and bolsters relevance of the model.

A direct determination of *in vivo* "serum survival phenotype", which likely would show serum resistance, would help to confirm the relevance of the model. These have been done using *N. gonorrhoeae* harvested directly from men with gonococcal urethritis (Nature 227, 382-384 (1970))—a project for the future!

Major comments

1. The use of chelators to isolate individual complement pathways appeared to indicate that the classical pathway was the only pathway involved in killing *H. pylori*. However, EGTA+Mg²⁺⁺ did not fully restore survival of organisms in the bactericidal assay, (---compared to survival in EGTA treated sera where the classical pathway is fully blocked); nor would it be expected to because the alternative pathway primarily serves to amplify the classical pathway (not active here because of chelation of Ca²⁺⁺) although direct activation of the alternative pathway has been suggested by others (*Helicobacter* 1 e, 28-33 (1996)) (zymosan is said to be the only unambiguous direct activator of the alternative pathway). Therefore the alternative pathway cannot be totally ruled out as participating in the role of C in killing *H. pylori*. Furthermore, EGTA alone diminishes free Mg²⁺⁺ but does not eliminate it, possibly permitting expression of some alternative pathway activity.

The use of C2 deficient sera (alternative pathway intact) and Factor B (FB) deficient sera or anti-FB reagent (classical pathway intact)—all commercially available, permit a cleaner separation and avoids the effect of chelators on the "health" of the organisms in the bactericidal assays, which can be relevant in the interpretation of results of bactericidal assays. Divalent cations help to maintain integrity of bacterial outer membranes. The killing assays should be repeated using these biologic reagents to corroborate assays that contain chelators. .

2. The bactericidal assays do not indicate the numbers of organism used in the assays; only that the ODs were 0.1; the bactericidal reaction mixtures were diluted before they were plated implying that the reaction mixtures likely had several logs of organisms, yet the displayed data used percentage survival instead of increases/decreases of log amounts—the usual way of describing this type of assay. It is unclear how the authors go from logs of organism in the reaction mixtures to percentage survival in the displayed data! A derivation of these calculations should be described in greater detail for clarification.

Minor comments

1. The first sentence in the Abstract is incorrect. Studies of complement interactions with infectious agents have been studied in other than bloodstream infections; the authors should look

these up and modify their statement.

2. Is *H. pylori* infection-free NHS presumed to be free of antibody directed against the organism? What is thought to be the basis for serum-sensitivity of *H. pylori*:

- antibody-free bactericidal activity due to direct activation of the classical pathway of C by *H. pylori* (I do not know of any precedent here)
- "natural antibody" in NHS that activates via the classical pathway of C (consistent with opinions about serum-sensitivity of other bacteria).

REVIEWER COMMENTS

Reviewer #1 (Remarks to the Author):

In this manuscript, Hu et al. present data that provide important additions to the literature on the complement system and *H. pylori*. The manuscript is well written and presented in a logical manner with well thought out schematics. Using a combination of mouse models, genetic manipulation of *H. pylori*, as well as altering culture conditions, the authors show that complement attenuates gastric colonization by *H. pylori* through the classical/lectin pathway. *H. pylori* can utilize host L-lactate to resist complement by blocking the accumulation of C4b, a cleavage product of C4, on the *H. pylori* surface. The following are suggestions to improve this study.

1. While this manuscript presents interesting and important results, with experiments that are well conducted, and data analysis that is well performed, it is lacking in supporting data of the human relevance. Could the authors look at complement in human sera from patients with and without *H. pylori* infection?

RESPONSE: Thank you for bringing this point to our attention, that the connection to human relevance needs to be more clearly addressed. Complement in human patients and *H. pylori* had been studied previously and we've revised the introduction to make this more clear. Specifically, Berstad and colleagues found that *H. pylori* was sensitive to human complement, even with serum from healthy human volunteers that were negative for IgG and IgA to *H. pylori*. Furthermore, they found that activated C3 was present at the surface of the human gastric epithelium, significantly more often from patients with than without *H. pylori* infection, suggesting that complement is found in the human stomach and *H. pylori* triggers complement activation there (Berstad *et al.*, 1997; Berstad *et al.*, 2001). These studies provide evidence that complement is relevant during *H. pylori* human gastric colonization. This information is now in the introduction Line 58-62: "Previous observations have suggested there is an interaction between *H. pylori* and human complement. For example, *H. pylori* triggers activation of human complement in vitro²⁸. Furthermore, *H. pylori*-infected patients exhibit significantly higher levels of the C3 complement protein in gastric samples as compared to *H. pylori* negative ones⁸", and at lines Line 67-69: "Thus, the relevance between *H. pylori* infection and gastric complement activation has been suggested but the features of these interactions have not yet been elucidated".

References:

Berstad, A. E., Brandtzaeg, P., Stave, R., & Halstensen, T. S. (1997). Epithelium related deposition of activated complement in Helicobacter pylori associated gastritis. *Gut*, 40(2), 196-203.

Berstad, A. E., Høgåsen, K., Bukholm, G., Moran, A. P., & Brandtzaeg, P. (2001). Complement activation directly induced by Helicobacter pylori. *Gastroenterology*, 120(5), 1108-1116.

2. The authors looked at a 2-week *H. pylori*-infection and determined that complement restricts colonization of *H. pylori*. Does this phenotype in colonization persist at later time points of infection?

RESPONSE: Excellent suggestion, and we recently completed a one-year infection so have been able to add this later-stage infection data. These data further support the conclusion that complement resistance is required for *H. pylori* gastric colonization even at late stages. This data has now been added to a revised Fig. 5, as panel 5B, and described in Results Line 240-244, as "Because *H. pylori* creates chronic infections, we assessed how long-term loss of complement would affect *H. pylori* colonization. We allowed *H. pylori* WT and $\Delta lctP$ mutant infections in both WT and C3^{-/-} mice to persist for one year. In this case, we found that loss of C3 allowed sustained recovery of the *lctP* mutants to WT levels (Fig. 5B), similar to the outcome after two weeks".

The updated Fig. 5B with figure legend (Line 746-755):

Figure 5. Loss of *lctP* renders *H. pylori* colonization deficient in a manner that is recovered by loss of complement

Groups of C57BL/6J WT and isogenic $C3^{-/-}$ mice were orally infected with the *H. pylori* PMSS1 WT or $\Delta lctP$ strains for either (A) 2-weeks or (B) 1-year. The WT and $C3^{-/-}$ mice were co-housed at weaning for at least 4-weeks before the infection and remained co-housed for the whole experimental period. At the indicated time points, the stomachs were collected and bacterial CFUs determined by plating. Each point represents one mouse sample, with bars representing the mean. Gray dotted lines represent the limit of detection. The p-values were obtained by two-way ANOVA (Tukey's test). The significance was indicated as * ($P < 0.05$), ** ($P < 0.01$), and n.s. (no significance).

3. The authors show bacterial load in the $C3^{-/-}$ mice corpus is 100-times that of the load in wild-type mice. Is the pH in the gastric corpus of both mice equivalent?

RESPONSE: Thank you for raising this important question, since pH is a critical strategy of the host to restrict bacteria in the gastric region. To address this question, we've now measured pH in both wild-type and $C3^{-/-}$ mice, also comparing either *H. pylori* infection-free or infected conditions. The pH in the corpus was lower compared to that of the antrum of each mouse type, but there were no differences between wild-type and $C3^{-/-}$ mice in either the corpus or antrum, either free or with *H. pylori* infection. This result has now been included in the main text as a new panel in Fig. 1, Fig. 1F. It is described in the results at Line 103-106: "As gastric acid, mainly produced in the corpus, is an important antimicrobial host factor in the GI tract, we examined whether $C3^{-/-}$ mice have differences in gastric pH compared to WT mice and detected no differences between mice strains, or with and without infection (Fig. 1F)".

Figure 1F:

Figure legend (Fig. 1F) is updated in Line 685-686: “Gastric pH levels in either antrum or corpus region of both WT and C3^{-/-} mice, in *H. pylori* free or infected condition”.

The methods for pH measurements are also added to the Methods at Line 433-438 as follows:

Mouse gastric pH measurement

The pH measurement was performed immediately after stomach dissection. The forestomach region was removed to allow the electrode of pH meter to enter the interior regions of stomach and closely touch the inner wall of either corpus or antrum subregion. The pH value was determined using pre-calibrated portable pH meter LAQUA with a 9618S-10D Micro ToupH electrode (3mm diameter) (HORIBA, Ltd., Kyoto, Japan).

4. How was inflammation affected in the gastric antrum and corpus of WT mice and C3^{-/-} mice with and without *H. pylori* infection? It has been shown that lower levels of bacterial colonization can be correlated with increased gastric inflammation.

RESPONSE: Thank you raising this point, as it warrants more explanation. As the reviewer states, inflammation has been shown to impact final bacterial levels, but in later infection and not at the two week time point. Indeed, that is one reason we focused on two week infections because previous work has shown there is minimal recruitment of immune cells at this time point (Arnold *et al.*, 2017), and we reasoned complement can be studied without a large effect of these additional variables. Specifically, Arnold *et al.* (2017) found that adult wild-type mice possess low levels of immune cells in the stomach at the basal state, including dendritic cells and macrophages. Only after three weeks of *H. pylori* infection, do cell numbers rise above the basal state for neutrophils, macrophages, MHCII⁺ monocytes and CD4⁺ T cells, inducing a mild level of inflammation. This work was done with *H. pylori* strain PMSS1, like our work, and the same mouse model, so these findings are directly applicable to our study. It is possible, however, that there is a low level enhancement of inflammation in the C3^{-/-} mice, since C3 absence causes complement dysfunction, including blockage of anaphylatoxin release (C3a and C5a) and lowered membrane attack unit formation (Gutzmer *et al.*, 2004; Aksamit *et al.*, 1981; Ehrengruber *et al.*, 1994). Combined, these studies suggest that inflammation plays minimal roles at the two-weeks infection time point, but there may be subtle effects that we cannot rule out or detect. We emphasized these ideas in the discussion at Lines 290-295, as “We propose that cell lysis dominates the complement mediated activities in vivo because there are few immune cells present in the stomach during *H. pylori* early infection²⁷. Indeed, immune cells are only recruited above basal levels after ~ 3 weeks of infection²⁷. There may be subtle effects that we cannot rule out or detect, however, due to low level immune cell recruitment and effects of C3 loss on anaphylatoxin release (C3a and C5a)^{2,3,4}. Future work will be required to tease out the precise roles of complement in the stomach”.

References:

Arnold, I. C. *et al.* NLRP3 controls the development of gastrointestinal CD11B⁺ dendritic cells in the steady state and during chronic bacterial infection. *Cell Reports* **21**, 3860–3872 (2017).

Gutzmer, R., Lisewski, M., Zwirner, J., Mommert, S., Diesel, C., Wittmann, M., ... & Werfel, T. (2004). Human monocyte-derived dendritic cells are chemoattracted to C3a after up-regulation of the C3a receptor with interferons. *Immunology*, *111*(4), 435-443.

Aksamit RR, Falk W, Leonard EJ. Chemotaxis by mouse macrophage cell lines. *J Immunol* (1981) **1950**(126):2194–9.

Ehrengruber MU, Geiser T, Deranleau DA. Activation of human neutrophils by C3a and C5a. Comparison of the effects on shape changes, chemotaxis, secretion, and respiratory burst. *FEBS Lett* (1994) **346**:181–4. doi:10.1016/0014-5793(94)00463-3

Reviewer #2 (Remarks to the Author):

Review for Hu and Ottemann, 2022, Nature communications

1. Using a combination of *in vivo* and *in vitro* methods, Hu and Ottemann present an exciting discovery that *H. pylori* colonisation and complement resistance is significantly affected by L-lactate uptake and interestingly, that complement seems to particularly affect colonisation in the corpus region of the stomach, compared with the antrum. The data supports a novel mechanism of complement resistance in which lactate uptake decreases C4b deposition on the bacterial cell surface. The mechanism for how lactate enables *H. pylori* to modify levels of C4b on the cell surface remains to be fully characterised and the authors speculate on potential mechanisms in the discussion. Overall, the data fully supports the authors conclusions, but clarification regarding Figure 4 should be included (see comments below). The authors show that this mechanism is not triggered by the alternative pathway, yet the authors do not investigate further whether lactate could be affecting the classical or lectin pathway. A suggestion to investigate this is suggested below, although it is recognised that fully characterising this may be outside the scope of this study. In summary, this paper represents a shift in our understanding of the role of complement in *H. pylori* colonisation and complement resistance. All methodology used is sound and reproducible. A limitation of this study is that all data is collected in a mouse model of infection, rather than any investigations in human samples. Have the authors considered quantifying the lactate using human gastric corpus and antrum tissue as an initial rationale for these findings being relevant to the human system? I hope that the comments below will be useful in improving the manuscript.

RESPONSE: Thank you for this important comment, which is similar to that of reviewer 1 point 1, so please see our response there discussing complement in human infections.

2. Line 57, Typo, worldwide to worldwise

RESPONSE: Changed

3. Line 80, italicise ‘*in vivo*’ if this adheres to journal formatting.

RESPONSE: Changed

4. Line 90, C57Bl/6 conventional naming, inconsistent- Figure 1, panel B uses C57BL/6J and line 598 uses C57Bl/6 please use standard format and ensure consistency throughout.

RESPONSE: Changed to C57BL/6J and double checked throughout

5. Line 97, Gland counting – A supplementary figure showing a representative microscopy image of gland occupancy versus gland load is very important here for the reader to understand the analysis and the clarity of images you are using to establish quantification of these parameters. Have you got data to support that the plasmid expressing GFP is stable and maintained during infection? If so, this would be important to mention here.

RESPONSE: Thank you for this good suggestion, and we agree that the figure is useful—the experimental approach is shown in Fig. 1B and we rewrote the text to make sure this was clearly

indicated at Line 101-102 (“Colonization in the important gastric sub-region, the gastric glands was quantified as exhibited in Fig. 1B,...”). We agree with the reviewer that example gland images would be useful, and have added them to Fig 1B as well. To understand the plasmid stability, our lab has extensively characterized this GFP-carried plasmid and determined that it is stably maintained during infection in Keilberg *et al.* (2016). This information has been added in the methods at Line 370-374, as “For mouse model infections, GFP-expressing PMSS1 WT and $\Delta lctP$ strains were constructed by natural transformation with plasmid pTM115²⁵ to create PMSS1 pTM115 (KO1568) and PMSS1 $\Delta lctP$ pTM115 (KO1711). pTM115 has previously been shown to be stably maintained during infection with robust high level GFP expression²⁵”.

Reference:

25. Keilberg, D., Zavros, Y., Shepherd, B., Salama, N. R., & Ottemann, K. M. (2016). Spatial and temporal shifts in bacterial biogeography and gland occupation during the development of a chronic infection. *MBio*, 7(5), e01705-16.

6. Line 104/105- during ‘the’ infection period used here. Could you define this in brackets ie (14 days)?

RESPONSE: Thank you for pointing out this lack of clarity. It is a 14 day infection period. We added a note to make this infection period clear in current Line 112-113, as “studies have shown that during infection period used here (14 days)”.

7. Figure 1- intriguing observation re corpus and antrum- have the authors attempted to quantify the levels of C3 (and/or alternative complement antibody components) in a WT infected stomach perhaps by flow cytometry to understand if the quantities differ between regions, which could complement the lactate quantification findings to explain the observed phenotype?

RESPONSE: We agree with the reviewer, that there may be differences in complement component levels between regions of the stomach. This idea is mentioned in the original discussion (“This bacterial load difference could be due to higher complement levels in the corpus, or alternatively, higher complement resistance in the antrum”). We have done experiments to monitor the levels of several complement components in different regions of stomach using western blotting. We found that different components exhibited varied levels. Some components were higher in corpus than the antrum, while others were at similar levels between corpus and antrum (data not shown). Such complexity was also reflected in a previous study that examined the levels of complement components using gastric samples of human patients, where they reported that C3 was more abundant in antrum of gastritis patients, while membrane attack complex, also called terminal complex (TCC), was not different between antrum and corpus (Berstad *et al.*, 1997). Both our observations and published work thus suggest that it is difficult to estimate complement activity based on the level of individual components. For a complete picture, all complement components would need to be quantified. Because complement components are highly regulated, however, we are not sure whether simple levels would reveal a full story. For these reasons, we feel determining whether there is higher complement activity in the corpus is beyond the scope of this work.

We also presented the idea that the differences in complement response may be due to different lactate levels between the two regions. Indeed, since there was a higher level of lactate in the antrum, we speculated in the original *that the defect of H. pylori corpus colonization was due to a slow induction of lactate* (Fig. 1D and new Fig. 1G, original supplement Fig. 2).

To capture these ideas, we’ve modified the discussion at Line 273-280, about whether the differences are due to lactate or complement levels as “It is complicated to decipher the basis of this phenomenon from the aspect of investigating the prevalence of subregional complement components. Previous work has

examined the levels of key complement components in the stomach, and found they exhibited varied patterns between antrum and corpus, with activated C3 more abundant in antrum and the membrane attack complex deposited equally between the two regions⁸. We noticed such complexity as well from examining mouse samples. In future studies, a systematic analysis would be needed to elucidate whether there is regional variation in complement activities and how this differentially impacts local *H. pylori* viability”.

Reference:

Berstad, A. E., Brandtzaeg, P., Stave, R., & Halstensen, T. S. (1997). Epithelium related deposition of activated complement in Helicobacter pylori associated gastritis. *Gut*, 40(2), 196-203.

8. Figure 1 title- plays ‘a’ vital role instead?

RESPONSE: Fixed.

9. Line 607- error bars (E, F). There is no panel F in Figure 1- I think this is meant to say (D, E)

RESPONSE: Corrected Line 688 to say (D, E) as pointed out.

10. Figure 2, what percentage of NHS is used in Panel B? Assumption is that this is 10% but this should be clearly defined in the legend or figure itself. Also, the line surrounding the panels should be removed.

RESPONSE: Thank you for the comment, we re-labeled the percent of serum to be more clear. The update is “All tests were carried out with 10% serum”, in Line 704.

11. Line 619- add ‘respectively’ after the word activation.

RESPONSE: Done

12. Figure 3C- it is not clear to me why the authors refer the reader to panel Figure 2C to visualise the survival in 10% WT mouse serum (22%) and compare this to survival shown in Figure 3C (80%). It would be much more logical for the data of survival in 10% WT mouse serum to be a bar in Figure 3C (ie 10% mouse serum, minus L-lactate), so that a direct comparison can be visualised, without switching between figures.

RESPONSE: Thank you for your comment. We revised the presentation of the mouse data in Fig. 3C so that all comparisons are within the panel—10% mouse serum, ± lactate.

13. Figure 3G- it would be logical to add a WT column to this graph such that survival in the WT strain can be directly compared statistically to survival in the Δ lct strain in NHS.

RESPONSE: Thank you for this suggestion. We combined the original Fig. 3A and 3G as the new Fig. 3A to be able to make relative comparison.

14. Figure Supp 2. Perhaps the x-axis could be labelled more clearly to indicate more time points during the 21 hour period.

RESPONSE Thank you for this suggestion, we have labeled more time points on the x-axis to make the time course clearer.

15. Figure 4B, labelling typo, ‘H. pylori stains’

RESPONSE: Thank you for the comment. We have corrected it.

16. Figure 4B and 4C. The results comparing total C4b accumulation seems to be telling different stories between panel 4B and 4C. When comparing C4b accumulation in panel B, there seems to be little difference between +/- lactate after 1 minute incubation and the authors therefore conclude, lactate has no impact on the accumulation of C4b. However, when looking at Panel C, +/- lactate at 3 and 30 minutes, there seems to be much less C4b in the presence of lactate versus the absence. Moreover, the quantification data in the bottom of Panel 4C shows a significant difference between Total C4b +/- lactate at 3 and 30 minutes, so the authors should clarify their statement on lines 184-186, which reads as the following..... ‘C4b accumulation, however, was not different with or without lactate in WT *H. pylori*, nor did it change between WT and $\Delta lctP$ strains (Fig. 4B). This outcome suggests that lactate does not prevent C4b formation, and therefore 186 does not affect steps upstream of this event.’ Given that this data forms part of an important conclusion, I would emphasise that this conclusion should be modified given the differences in these time points.

RESPONSE: Thank you for these comments. We realized our presentation was not clear and caused some confusion, with regard to our statement that on original lines 205-207, “..... ‘C4b accumulation,

however, was not different with or without lactate in WT *H. pylori*, nor did it change between WT and Δ lctP strains (Fig. 4B)”, the use of the different time points in Fig. 4B and 4C.

First, our conclusion was reached from evaluating the total C4b formation after allowing the reaction to go to completion, at 60 minutes. To more clearly present this idea, we re-organized our description in Line 205-208, as “C4b accumulation at the end of the incubation period (60 minutes) was not different with or without lactate in WT *H. pylori*, nor did it change between WT and Δ lctP strains (Fig. 4B). This outcome suggests that lactate does not prevent C4b formation, and therefore does not affect steps upstream of this event.”

We did also notice the lower levels of C4b at one minute, and so we assessed additional time points in Fig. 4C to gain more insight into whether L-lactate may slow down C4b formation. Indeed, this seems to be the case so we described these ideas and results in more detail at Line 208-210 as “We did note that L-lactate may slow down C4b formation in *H. pylori*, as evidenced by the slower accumulation of total C4b at early time points (Fig. 4C, 4D)”.

17. Figure 4C – the bottom panel on this, ie the quantification data should be labelled as Figure 4D, which will make it easier to reference to in the text.

RESPONSE: Thank you for the suggestion, we agree that making this Fig. 4D is more clear. As you suggested, we labeled the quantification data with Fig. 4D.

18. Figure 4 legend- the conditions 2-3 for Panel C are labelled in a different order to how they appear in the gel. Please correct in the figure legend.

RESPONSE: Thank you for the suggestion and sorry for that confusion. As you suggested, we have reordered the description to match gel panel appearance order.

19. Typo, Figure 4 legend, ‘quantification purpose’ should read ‘purposes’

RESPONSE: Thank you for the suggestion. We revised this description as described in Reviewer 2, #16 and #20, and this part is removed.

20. Figure 4 – the y axis on the Panel C graph states ‘Ratio of ...’. How has this ration been calculated? In relation to what? Can this be clarified in the legend.

RESPONSE: Thank you for pointing out this lack of clarity. We’ve modified the figure legend to be clearer as described at Line 740-742, as “Bands were quantified by normalizing each band to the amount of total C4b or bound C4b at 3 minutes (leftmost band in each treatment group) using the Bio-Rad Image Lab software”.

21. Line 196-198 states that C4c formation was undetectable. I would suggest that a positive control band of C4c detection is required here (even if this is a supplementary figure) in order to justify that if C4c were present, the antibody would detect this and therefore fully support the conclusion that Factor I and C4Bp are not acting in this pathway.

RESPONSE: Thank you for this important suggestion. We added the positive control data by using human IgG as a classical complement pathway activator, as used previously (Berstad *et al.*, 2001). The updated Fig. 4B shows that C4c can now be detected at 60 min post activation, indicating that our system can generate and recognize C4c, but that C4c is not created by *H. pylori*. We have modified the results

text at Line 220-221 with “C4c was detectable in the positive control samples, but nearly undetectable ...”. The method is updated at Line 461-463 (“As positive control for complement classical pathway activation and the creation of C4c, human IgG (Invitrogen) was heated at 63°C for 20 min to trigger aggregation. 1mg/ml heated IgG (final concentration) was co-incubated with NHS to activate complement²⁸”). And in the Fig. 4 legend, the new updated Fig. 4B at Line 732-733: “A positive control of complement activation and C4c formation was triggered by heated human IgG”.

Updated Fig. 4B:

Reference:

Berstad, A. E., Høgåsen, K., Bukholm, G., Moran, A. P. & Brandtzaeg, P. Complement activation directly induced by *Helicobacter pylori*. *Gastroenterology* **120**, 1108–1116 (2001).

22. Figure 5. All comparisons of significance seem to be made aside from the colonisation densities in the antrum of the Δlct strain in the WT mouse compared to the C3 knockout mouse. This should be compared for completeness.

RESPONSE: Thank you for the comments. We updated the full set of comparison labels in the updated Fig. 5A (original Fig. 5) and also included full statistical analysis of the newly made Fig. 5B (*H. pylori* chronic colonization).

Fig. 5A:

23. Lines 219- the statement that co-housing enables mixing of the microbiota, can a study be referenced to support this suggestion that 4 weeks leads to changes in microbiota of co-housed mice. I am assuming this data may come from the lower gut rather than the stomach? This should be clarified given that no data is shown on the microbiota compositions between these mice. Currently, I do not see any paper to talk about.

RESPONSE: Thank you for highlighting that this section needs clarification. As you’ve pointed out, we based our experimental design for co-housing mice on previous intestinal microbiota studies, and the

reviewer is correct that there has been no direct examination of gastric microbiota composition mixing with co-housing. However, it is known that mice are coprophagic, that the gastric microbiome is influenced by fecal consumption, and that mice within a cage group share similar intestinal microbiomes (Caruso *et al.*, 2019). We thus were inspired by these studies and from published intestinal studies to derive the co-housing method used here. As it is known that the GI tract microbiota of new born mice is not fully established until after weaning (Caruso *et al.*, 2019), we co-housed WT and C3^{-/-} mice at weaning to allow mixing at this critical stage. We have updated our description to more clearly present these ideas in Line 244-247, as “To minimize any gastric microbiota differences between WT and C3^{-/-} mice, we cohoused these two types of mice starting at weaning, when the microbiota is not established, and throughout the infection period. This type of approach has been demonstrated to promote the equalization of intestinal microbiota⁶⁸”.

Reference:

68. Caruso, R., Ono, M., Bunker, M. E., Núñez, G. & Inohara, N. Dynamic and asymmetric changes of the microbial communities after cohousing in laboratory mice. *Cell Reports* (2019).

24. Line 236, Typo ‘tracks’

RESPONSE: Fixed

25. Line 245, Typo ‘elevated a L-lactate’

RESPONSE: Fixed

26. Line 254, the statement that the C4b deposition resistance mechanism has not been reported for bacteria- can this be expanded to include information on whether the mechanism of lactate-mediated complement resistance in other bacteria (apart from *Neisseria* mentioned in the introduction) has been defined or not, or a comment as to how widespread this mechanism could be across bacteria?

RESPONSE: Thank you for this important point to discuss how our work relates to other studies on lactate utilization and complement. Aside from the *Neisseria* work, there have been no other reports connecting lactate and complement. We modified our discussion in Line 300-305 to make this clear as “The complement resistance mechanism described here, lactate promoted bacterial-intrinsic C4b removal, has not been reported previously. Whether it is widespread remains to be seen, as there have been very few studies on the interplays between microbes and complement in the GI tract. Whether there are connections to lactate is also a new area, although several recent studies have identified lactate as a critical nutrient to promote bacterial expansion in GI tract⁶⁶”.

New added reference in manuscript:

66. Gillis, C. C. et al. Dysbiosis-associated change in host metabolism generates lactate to support salmonella growth. *Cell Host & Microbe* 23, 570 (2018).

27. Line 273, Typo ‘an colleagues’

RESPONSE: Fixed

28. Line 355, can justification be given as to how infection-free status was confirmed in individuals from which the NHS was isolated.

RESPONSE: This NHS was purchased from a commercial supplier, and is evaluated based on *H. pylori* antibodies. We updated the methods as to make this method clear: “Normal human serum (NHS, Sigma), negative for anti-*H. pylori* antibodies as indicated by manufacturer, was used for *H. pylori* serum sensitivity tests.” in Line 392-393.

29. Although the authors correctly conclude that C4b deposition is significantly decreases by lactate, is there a possibility that *H. pylori* could be regulating the activity of MASPs in the lectin pathway and therefore the cleavage of C4 and binding of C4b to the cell surface? Can this be explored by differentiating between whether this mechanism is triggered by the antibody or lectin pathways? Perhaps the NHS could be depleted of antibody and determine sensitivities to complement +/- Lactate?

RESPONSE: This is a great suggestion, and addressing it allowed us to gain a clearer view about how lactate mediates C4b destabilization from the *H. pylori* surface, including stronger evidence that the lectin pathway is not involved. Based on the current view about classical and lectin pathway activation, they share the same downstream reactions after C1 complex and MASP formation to induce C4 activation. Both C1 complex and MASP target the same peptide bond cutting site on C4- α chain to generate C4a and C4b (Fig. 4A).

To better understand the potential role of classical and lectin pathway activation on C4 activation and C4 deposition on *H. pylori* surface, we've added experiments to pinpoint whether *H. pylori* was equally targeted by classical and lectin pathway. Initially we used serum treated with EGTA/ Ca^{2+} / Mg^{2+} , and we've now added experiments with C1q- and C2-depleted human serum to treat *H. pylori*. C1q-depleted serum blocks activation of the classical pathway, leaving both lectin and alternative pathways on, while C2-depleted serum blocks activation of both classical and lectin pathways, leaving the alternative pathway on. *H. pylori* was not sensitive to C2-depleted serum, agreeing with the observation from EGTA/ Mg^{2+} conditioned-serum treatment. We further found that C1q-depleted serum lost its ability to kill *H. pylori*, suggesting that *H. pylori* was not sensitive to either the lectin or alternative pathway. This information is included in the results and in updated Fig. 2B. Based on these combined observations, C4 activation is likely triggered through the classical pathway, and the lectin pathway plays only a very minor role.

We've updated the description of these new observations in Line 136-152, as “Complement can be activated through three pathways, classical, lectin, or alternative. These pathways can be blocked by treating with EGTA, and then differentially re-activated by adding back either Ca^{2+} , to activate the classical and lectin pathways, or Mg^{2+} , to activate the alternative pathway. We used this differentially-activated serum to evaluate which pathway targeted *H. pylori*. The presence of EGTA completely abrogated the susceptibility of *H. pylori* to complement killing (Fig. 2C). Ca^{2+} supplementation resulted in the killing of *H. pylori* to levels that were nearly identical to those in untreated NHS, suggesting that *H. pylori* is mostly targeted by the classical and/or lectin pathways (Fig. 2C). This observation was confirmed using Factor-B depleted serum, which loses the alternative pathway (Fig. 2C). To further distinguish the role of the classical from the lectin pathway, C1q-depleted serum was applied to exclusively block classical pathway activation², and had no obvious effect on bacterial viability, suggesting that the classical pathway played a critical role while the lectin pathway played a minor role in *H. pylori* elimination (Fig. 2C). Meanwhile, adding back Mg^{2+} to EGTA treated NHS did not fully restore the killing capacity of the NHS, nor was C2-depleted serum able to impact bacterial viability, suggesting that alternative pathway played a minor role in *H. pylori* killing (Fig. 2C). Finally, killing assays with a higher percent of NHS that lacked specific pathways confirmed that the classical pathway played a central role in killing *H. pylori* (Supplemental Fig. 1)”.

Updated Fig. 2C and figure legend at Line 698-704: “(C) *H. pylori* cultures were treated by EGTA untreated NHS (grey bar, all pathways on); EGTA-conditioned NHS to block all complement pathways (white bar, all pathways off); EGTA + Ca²⁺ (blue bar, classical and lectin pathway on); EGTA + Mg²⁺ (green bar, alternative pathway on); C1q-depleted human serum (yellow bar, alternative and lectin pathway on); C2-depleted human serum (light orange bar, alternative pathway on only); or Factor B-depleted human serum (light red bar, classical and lectin pathway on). All tests were carried out with 10% serum.”

Updated Supplemental Fig. 1 and figure legend:

Supplemental Figure 1 L-lactate protects *H. pylori* from complement mediated elimination

Overnight cultures of *H. pylori* PMSS1 WT, with or without L-lactate supplementation, were treated by inactive and active human serum at 30% final concentration for one hour, and then viable bacterial CFUs were determined by plating. The survival percent was determined as described in Fig. 2. In the panel, *H. pylori* cultures were treated by NHS only (grey bar), EGTA-conditioned NHS (white bar), C1q-depleted human serum (yellow bar), C2-depleted human serum (orange bar), or EGTA plus Mg²⁺ (green bar) as in Fig. 2C. All tests were applied with 30% serum. The results were presented as survival percent \pm standard deviation (SD), and were derived from triplicate biological samples with a triplicate of each given sample. The *p*-values were obtained with a two-tail student *t*-test. The significance is indicated as **** (P<0.0001), or n.s. (not significant).

Reviewer #3 (Remarks to the Author):

The manuscript "Helicobacter pylori initiates successful gastric colonization by utilizing L-lactate to promote complement resistance" by Shuai Hu and Karen M. Ottemann describes that *H. pylori* is sensitive to complements, and in particular through the Ca²⁺-dependent classical pathway. *H. pylori* infect complement depleted mice in higher infectious loads. Uptake of L-lactate makes *H. pylori* more complement resistant by blocking deposition of the complement C4b on *H. pylori*. This is a story similar to the *Neisseria* bacteria that activates biosynthesis pathways by uptake of L-lactate for complement resistance. The authors have previously published that L-lactate is taken up by *H. pylori* in cultures and this promotes increased growth rate. Thus, it is likely that L-lactate regulates many biosynthesis pathways in *H. pylori* including the large family of membrane proteins and most likely also its LPS-composition. My main comment is about the validity of the complement system and resistance at lower pH, ie the acidic condition in the stomach which is very different from the buffered pH in circulation where most of the complement system activation takes place, something which I hope the authors can test and sort out.

RESPONSE : Thanks to the reviewer for this very clear summary. One correction is that we did not previously publish the studies that L-lactate was taken up by *H. pylori*, that was Iwatani *et al.* (2014). We've double checked the manuscript to make sure that is correctly cited. And yes, as you'll see below, we've tested complement activity at low pH and have added that to the manuscript.

Main Comments:

1. In line 69 ,the authors mention "in *Neisseria meningitidis* by activating a sialic acid biosynthesis pathway that protects against complement's action¹⁸." Have the authors considered to explore this mechanism for better understanding of the complement resistance and the protection against C4b accumulation on the *H. pylori* bacterial surfaces?

RESPONSE. This is an excellent suggestion, to examine whether *H. pylori* develops similar L-lactate triggered anti-complement strategies as does *N. meningitidis*. In fact, we started our work by examining whether *H. pylori* employs a *Neisseria*-like mechanism by checking the status of *H. pylori* LPS sialylation in response to lactate. We found, however, that LPS sialylation was not stimulated by lactate compared to non-lactate-grown *H. pylori*. This result suggested that the anti-complement mechanism employed by *H. pylori* was different from that of *N. meningitidis*. Because this was negative data and to keep the manuscript focused, we did not present this investigation. Bacterial sialylation is mainly involved in alternative complement pathway resistance, by recruiting complement regulator Factor H and Factor I to mediate C3b cleavage. Our studies, shown in Fig. 2 and summarized in the response to Reviewer 2, comment 29, demonstrate that the *H. pylori* is not sensitive to the alternative pathway.

We've improved our discussion of these differences of lactate mediated anti-complement mechanism between *N. meningitidis* and *H. pylori* in the discussion section at Line 333-337, as "*N. meningitidis* utilizes lactate to evade alternative pathway activation by stimulating the sialic acid biosynthesis pathway to robustly recruit Factor H that facilitates Factor I for C3b cleavage^{8,18}. In *H. pylori*, L-lactate was not found to promote *H. pylori* LPS sialylation from our preliminary investigation, but exclusively mediates classical pathway defense by promoting the removal of C4b".

2. Line 83, The authors state" Overall, our results reveal that complement plays an important role in controlling bacterial colonization in the GI tract" (and in the first sentence in the Discussion). But I do not think a result from the mouse stomach can be applied as a conceptual result for the entire GI tract, but suggest the authors to be a bit more modest and refer to the gastric tissue/location.

RESPONSE: Thank you for the comment to make our description more precise. We modified our description as you suggested, e.g. at current Line 86-87 of the introduction "...complement plays an important role in controlling bacterial colonization in the gastric region of GI tract" and in the discussion at Line 253, as "...the complement system plays an important role in gastric defense"

3. In line 91-92 the authors describe the length of the mice infection; "After two weeks, total bacterial counts and gland colonization were evaluated in each of the two main stomach regions, antrum and corpus". Considering that *H. pylori* infection are chronic and life-long in humans, it would be reasonable to infect the mice for a period that corresponds to a chronic infection, i.e., for a few months instead of an acute infection as performed here. A chronic infection might reveal conditions that would better reflect the human infection scenario.

RESPONSE: This is an important suggestion to examine whether complement plays roles in inhibiting *H. pylori* chronic colonization, similar to Reviewer 1, point 2. In that response, we describe our newly added one year infection data in Fig. 5B, and the finding that complement plays a role at these chronic time points.

4. In lines 94-95 the authors find that "Total bacterial load in the C3^{-/-} mice corpus was significantly elevated 100-fold compared to WT mice (Fig. 1C)." This suggest that the complements in the mouse stomach are more prevalent in the corpus region. The authors should probe the antrum vs corpus mucosa to test if there is a difference in complement levels and hence availability

RESPONSE: This is a very important comment and suggestion, and similar to Reviewer 2, #7. Please see that response about complement quantification.

5. From line 124 the authors test the complement activation by Ca for classical pathway and Mg to activate the alternative pathway and killing of *H. pylori*. And Hp is sensitive for killing by complement activation by Ca for classical pathway. This in vitro test looks convincing, but is performed in pH condition that are most compatible with human and mouse sera, i.e., in the environments where the complement system is supposedly active. It is well established that lower pH inactivates the complement system, see e.g. this reference from 2016 by Dantas E., et al; Low pH impairs complement-dependent cytotoxicity against IgG-coated target cells. *Oncotarget*. 2016; 7: 74203-74216. <https://www.oncotarget.com/article/12412/text/>. And thus reduction in complement activation takes place already at pH 5-6. Thus, the authors need show that the complement activated killing of *H. pylori* can take place also at lower pH. The mouse experiments indeed show that *H. pylori* manage accomplish higher infection loads (higher CFU) in the complement depleted mice. But similar to the argument about reduced pH in the human stomach, vs test by human sera at neutral pH, it is important to remember that mice do not demonstrate low pH in the stomach, but are much higher in stomach pH compared to the human stomach. Complement activation at reduced pH in the mouse could be tested by use of the INS-GAS mice (the Jim Fox model) where human gastrin drives the acid secretion and the mice demonstrate a more acidic and hence human like stomach pH. Thus, the in vitro tests and in addition the mouse tests would benefit from testing during conditions that mimic the acidity of the human acidic stomach mucosa i.e., where the *H. pylori* infection is located and less in contrast to the pH of the sera/blood in circulation.

RESPONSE: Thank you to the reviewer for raising several points related to stomach pH and complement. First, as suggested, we've added additional experiments to examine whether complement was effective at low pH. We exposed *H. pylori* to 10% or 30% NHS at pH 3.5, and found that the complement was still active to significantly decrease *H. pylori* numbers, killing 40% or 92% of *H. pylori* during 1h treatment (Updated Fig. 2B). While this is somewhat less than at pH 7 with 10% NHS, where 60% of *H. pylori* were killed (Fig. 2A), it's still substantial and actually more killing with 30% NHS at pH 3.5 vs. 7. These

results suggest that complement retains activity at low pH. This new data is included in the results at Line 121-125 as “Considering that *H. pylori* colonizes the stomach where it is acidic, we examined whether complement is functional in a low pH environment. We exposed *H. pylori* to 10% or 30% NHS at pH 3.5, and found that the complement was still active to significantly decrease *H. pylori* numbers, killing 40% or 92% of *H. pylori*, respectively (Fig. 2B). These results suggest that complement retains killing capacity at low pH.”

The reviewer also raises the concern that the mouse model has a higher gastric pH than in the human stomach, which may reduce the relevance of the observed complement activation from mouse infection model to the human stomach. *H. pylori* survives in a pH range 4-8.5 (Rektorschek, 1998), but only multiplies at pH 7. This suggests that *H. pylori* survives and grows in the local niches of the stomach that have near-neutral pH, which would be similar between mouse and human. Complement is functional in such environment, even as low as pH of 3.5 (Fig. 2B), suggesting it would be active in both human and mouse stomach environments where *H. pylori* lives. We also have added data, in response to Reviewer 1 #3, with pH measurements of WT and C3^{-/-} mouse corpus and antrum. These ideas have been added to the discussion, at line 283-288, as “Another key aspect of the corpus is its lower pH compared to the antrum. Our results suggest that complement retains activity at low pH. Furthermore, *H. pylori* resides and multiplies in the local niches where the pH is closer to neutral⁶⁷. Although the human stomach is more acidic than the mouse one, because *H. pylori* resides at the near neutral epithelial surface, this pH differences may not have an effect on complement functions in *H. pylori* colonizing niches and suggests that our findings would apply well to humans”.

Figure 2B:

Updated Reference:

67: Rektorschek, M., Weeks, D., Sachs, G. & Melchers, K. Influence of pH on metabolism and urease activity of *Helicobacter pylori*. *Gastroenterology* **115**, 628–641 (1998).

6. In line 144; the authors conclude that “Lactate protected *H. pylori* from the classical/lectin pathway, activated by Ca²⁺, significantly raising the survival percent from ~40% to ~85%, but not the alternative pathway, activated by Mg²⁺, which did not differ ± lactate (Fig. 3E).” and this conclusion was also reiterated in the section entitled “L-lactate prevents accumulation of C4b on the *H. pylori* surface.” And I agree to that with no lactate added the classical pathway seems to be the active pathway in killing *H. pylori*. But when lactate is added the Survival-percent increases from 40 to 80 % and with so high levels, it is not possible to see any effect of the EGTA +/- Ca or Mg treatment. And since the killing mechanism cannot be interpreted from these bars, I do not think the authors can conclude that the classical pathway is

active also during the conditions of lactate supplementation. Since there is less killing with lactate, the authors can rerun this experiment with higher levels of human sera to kill of more of *H. pylori* and hence reduce the Survival percent. A lower bar (survival) opens up for interpretation by use of the EGTA, i.e., if Ca is still necessary for complement killing of the *H. pylori* or if the lactate has induced another mechanism for resistance and survival.

RESPONSE. Thank you for raising this very good suggestion, which is similar to that of Reviewer 2, #29. Please see our discussion there of higher levels of human serum, and inclusion of new experiments that use serum that retains only specific complement pathways.

7. Same page, line 152, the authors deliver an interesting result "In uninfected WT or C3^{-/-} mice, L-lactate levels were nearly the same, at 5 $\mu\text{mol}/\text{gram}$ tissue, in both corpus and antrum (Supplemental Fig. 1). After two weeks of *H. pylori* infection, the levels were elevated 2-3-fold in the antrum and ~ 1.5 fold in the corpus (Supplemental Figure 1)." Thus, infection activates lactate production in the gastric tissue. I have not followed the papers on *H. pylori*/ lactate but could not find that such induction of lactate had been described previously. Please explain how come you present this result in the Supplements and not in the main Figures. Please also present the results in the context of what has been previously described about infection-activated lactate production or if this is a novel result? A better context would help me and other readers to better understand the novelty of the presented results.

RESPONSE: Thank you for this suggestion, to include the lactate production in the main text. This has now been done and updated in new Fig. 1G and 1H. Lactate over production has been reported in several microbial infections, but the reviewer is correct that its production status during *H. pylori* infection has not been reported. Lactate induction has been reported to be vital for infection by some bacterial pathogens, such as *Salmonella* (Gillis *et al.*, 2018). There were suggestions that *H. pylori* might also influence tissue lactate. For example, *H. pylori* virulence factors CagA and VacA are able to upregulate gene expression of key glycolytic enzymes, through manipulating host cellular signaling pathways. To more clearly describe that the gastric lactate induction is new, we updated the results description at Line 170-172, as "Since lactate induction was not yet been reported during *H. pylori* gastric infection, we evaluated whether that L-lactate is present in the mouse stomach and how levels are affected by infection,..."

Reference:

Gillis, C. C. *et al.* Dysbiosis-associated change in host metabolism generates lactate to support salmonella growth. *Cell Host & Microbe* **23**, 570 (2018).

8. The induced lactate in the infected tissue would also have been very interesting to study also during chronic infection, such as a few months of infection instead of the short acute infection model used by the authors.

RESPONSE: Thank you for the suggestion. In this study we have not had a chance to thoroughly characterize lactate levels in chronic infection. We are currently performing an *H. pylori* chronic infection experiment and will monitor lactate levels, as you suggest. There is suggestive data that gastric lactate levels will be elevated during *H. pylori* chronic colonization. It was reported that gastric lactate levels were significantly elevated in gastric carcinoma patients compared to control subjects, as well as an increasing trend in gastric ulcer group (Piper *et al.*, 1970). Considering that *H. pylori* is one of the main risk factors of gastric cancer and gastric ulcer, lactate acidosis is likely to be associated to *H. pylori* chronic colonization.

Reference:

Piper, D. W., Kemp, M. L., Fenton, B. H., Croydon, M. J., & Clarke, A. D. (1970). Gastric juice lactic acidosis in the presence of gastric carcinoma. *Gastroenterology*, 58(6), 766-771.

9. Page 9, line 195, “To determine whether this pathway was operating, we monitored C4c formation, recognized by our anti-C4 antibody (Fig. 4B). C4c was nearly undetectable in either lactate-grown or non-lactate-grown WT and Δ lctP *H. pylori* (Fig. 4B), suggesting Factor I and C4BP were not acting.” This is of course an interesting result, but since the experimental systems are difficult, this result would warrant a positive control, i.e., the authors need show that the mechanism is indeed operational with the serum samples used for a bacteria that is well known to bind C4BP.

RESPONSE: Thank you for the comment, which is similar to that of Reviewer 2 #21. Please see our response there, where we describe new experimental data that included a positive control for C4c formation, in updated Fig. 4B.

10. In the final section “Complement resistance is critical for stomach colonization The authors describe the C57BL/6N (6N) mice whereas in the Fig 1 the authors use C57Bl6/J and in Suppl Fig 5 the mice are described as C57BL/6N mice and as C57Bl6/N mice. Would be good to see an explanation why the authors use different breeds of C57BL/6 (in addition, the nomenclature is C57BL/6N or C57BL/6J, not C57Bl6/N or C57Bl6/J).

RESPONSE: Thank you for pointing out that this needs clarification. The C57BL/6N mouse model was used to confirm the observed colonization defect of *H. pylori* Δ lctP mutant in C57BL/6J mice. Through the use of the C57BL/6N mouse model infection, we were able to rule out the possibility that mouse type plays a role to cause a failed infection of *H. pylori* Δ lctP mutant. Using C57BL/6J and C57BL/6N mouse model together suggested that LctP was required for *H. pylori* gastric colonization.

To more clearly present our ideas, we updated the description as “We confirmed this defective phenotype in another commonly used WT mouse model, C57BL/6N mice^{25,38}” in Line 233-234.

11. In the Discussion, line 261, “*H. pylori* virulence factor SabA is able to bind sialic acid, which facilitates blocking alternative pathway activation^{41,42}. I find this sentence /statement difficult to understand, which blocking? and which complement? pathway activation?”, please explain how come you find SabA of importance in this perspective.

RESPONSE: Thank you for pointing out that this part was not clear. We intended to use the SabA study to indicate that *H. pylori* is not sensitive to alternative pathway activation possibly because *H. pylori* has developed a strategy to resist the alternative pathway.

To more clearly describe the sialic acid mediated anti-alternative pathway mechanism, we modified our description as “The *H. pylori* virulence factor SabA binds sialic acid, which has high affinity to Factor H. Binding of Factor H allows *H. pylori* to activate Factor I for C3b cleavage, a process to block alternative pathway activation^{41,42}” in Line 311-313.

12. In the Discussion line 273, “Guillemin and colleagues found that the complement component C4 coding gene C4B, is upregulated upon *H. pylori* infection of gastric epithelial cells⁴⁷. I did not find this mentioned in Guillemin’s article, but maybe it is listed in a supplementary file? In that case the authors need point out where the reader can find this information.

RESPONSE. Good point. The information was mentioned in supporting Fig. 11 and supplemental table 4.

We've added this information as "...C4B, is upregulated upon *H. pylori* infection of gastric epithelial cells, as described in the Guillemin *et al.* (2002) Supporting Fig. 11 and Supplemental Table 4⁴⁷" in current Line 324-325.

Minor comments:

1. Page 3, line 51-52 describes the C3 to C3B whereas in Fig 1A. the C2 to C2B are illustrated. This goes also for Fig 6 which illustrates C2 but no C3. This needs some better coordination/ description to fit the text.

RESPONSE Thank you for the kind suggestion. To avoid the potential confusion, we modified our description in Line 49-52 for complement activation to better match the information of Figure 1A, as "In the classical and lectin pathways, a key step is the processing of C4 to its products C4a and C4b, which complex with activated C2 to catalyze the formation of C3 convertase. C3 convertase formation is a central step in the alternative pathway as well. Multiple proteolytic steps follow this central step to eventually result in activated products and MAC formation".

2. Line 57; "worldwise"

RESPONSE: Done

3. Line 64 the authors mention "complement evasion mechanisms unlike found in other microbial species¹⁴". I think it would be appropriate to describe these "other mechanism" to some extent either in the Introduction or in the Discussion. The Discussion is rather vivid and voluminous but does not explain complement resistance mechanisms in other well explored pathogens.

RESPONSE: Thank you for the comment. We've expanded our description of other complement mechanisms as suggested at Line 64-67, as "These known *H. pylori* resistance mechanisms rely on recruiting host proteins, but there are not any known *H. pylori* self-produced complement evasion mechanisms unlike found in other microbial species, such as *Pseudomonas* spp. and *Staphylococcus* spp. that produce various proteases to inactivate complement components¹⁴."

4. From line 124 the authors test the complement activation by Ca for classical pathway and Mg to activate the alternative pathway. But this is somewhat confusing since in the Figure 2 "H. pylori is sensitive to complement mediated elimination", the similar text can easily be misunderstood since Mg and Ca comes in the wrong order: "(B) Mg²⁺ or Ca²⁺ were individually added back to the culture to resume either the classical/lectin or alternative pathway activation".

RESPONSE: Corrected.

Reviewer #4 (Remarks to the Author):

General comments

The authors have developed a model that describes how *Helicobacter pylori* becomes resistant to complement (C) and therefore thrives in the stomach of experimental mice.

They show that *H. pylori* multiply in the corpus of the stomach when complement activation is truncated at the level of C3 (--tantamount to C resistance)

They also propose that, in vitro, *H. pylori* are susceptible to killing by C in normal human serum (NHS), exclusively by the classical pathway of C. Killing is reversed (nullified) by the presence of lactate

because of resultant disruption of the covalent binding of C4b to the surface of the bacteria, thereby halting progression of C activation through to insertion of MAC (C5b-9) that would result in killing of the organism. The evidence given for C4b disruption is solid!

The experiments are carefully performed and the use of the Δ lctP mutant ensures that effects of lactate are specific for their observed phenotypic effects on survival (“serum survival phenotype”) of WT *H. pylori* in serum. Lacking direct confirmation that *H. pylori* survive in vivo (in the mouse) and are resistant to serum killing in vivo because of lactate uptake, inoculation of mice with the Δ lctP mutant to demonstrate reduced or nonviability of this strain in vivo serves as a proxy (control) for in vivo specificity and bolsters relevance of the model.

A direct determination of in vivo “serum survival phenotype”, which likely would show serum resistance, would help to confirm the relevance of the model. These have been done using *N. gonorrhoeae* harvested directly from men with gonococcal urethritis (Nature 227, 382-384 (1970))—a project for the future!

Major comments

1. The use of chelators to isolate individual complement pathways appeared to indicate that the classical pathway was the only pathway involved in killing *H. pylori*. However, EGTA+Mg²⁺⁺ did not fully restore survival of organisms in the bactericidal assay, (--compared to survival in EGTA treated sera where the classical pathway is fully blocked); nor would it be expected to because the alternative pathway primarily serves to amplify the classical pathway (not active here because of chelation of Ca²⁺⁺) although direct activation of the alternative pathway has been suggested by others (*Helicobacter* 1 e, 28-33 (1996)) (zymosan is said to be the only unambiguous direct activator of the alternative pathway). Therefore the alternative pathway cannot be totally ruled out as participating in the role of C in killing *H. pylori*. Furthermore, EGTA alone diminishes free Mg²⁺⁺ but does not eliminate it, possibly permitting expression of some alternative pathway activity. The use of C2 deficient sera (alternative pathway intact) and Factor B (FB) deficient sera or anti-FB reagent (classical pathway intact)—all commercially available, permit a cleaner separation and avoids the effect of chelators on the “health” of the organisms in the bactericidal assays, which can be relevant in the interpretation of results of bactericidal assays. Divalent cations help to maintain integrity of bacterial outer membranes. The killing assays should be repeated using these biologic reagents to corroborate assays that contain chelators. .

RESPONSE: Thank you for the suggestion to use biological reagents to examine *H. pylori* sensitivity to individual complement pathway. As you suggested, we’ve now included data for specific complement component- (C1q, C2, or Factor B) depleted serum and increased the percent of serum as well to gain a cleaner separation of the activation of complement pathways and better determine the sensitivity of *H. pylori* to specific pathway activation. This comment is similar to that of Reviewer 2 #29 and Reviewer 3 #6, so please see the detailed response at Reviewer 2 #29.

2. The bactericidal assays do not indicate the numbers of organism used in the assays; only that the ODs were 0.1; the bactericidal reaction mixtures were diluted before they were plated implying that the reaction mixtures likely had several logs of organisms, yet the displayed data used percentage survival instead of increases/decreases of log amounts—the usual way of describing this type of assay. It is unclear how the authors go from logs of organism in the reaction mixtures to percentage survival in the displayed data! A derivation of these calculations should be described in greater detail for clarification.

RESPONSE: We updated the method for bacterial survival ratio with more clear description, at Line 395 to include the actual bacterial number “...diluted to OD₆₀₀ of 0.1 (~10⁷ bacteria)” and to describe how we calculated the percentage survival in Line 402-404: “The bacterial survival ratio was calculated by

determining the number of colony forming units (CFUs) after exposure to either active NHS or to inactive NHS, and then dividing the numbers in the active serum by those in the inactive.”

Minor comments

1. The first sentence in the Abstract is incorrect. Studies of complement interactions with infectious agents have been studied in other than bloodstream infections; the authors should look these up and modify their statement.

RESPONSE: Thank you for bringing this to our attention. We modified the first sentence of the abstract to “The complement system has long been appreciated for its role in bloodborne infections, but its activities in other places, including the gastrointestinal tract, remain elusive”.

2. Is *H. pylori* infection-free NHS presumed to be free of antibody directed against the organism? What is thought to be the basis for serum-sensitivity of *H. pylori*:

- antibody-free bactericidal activity due to direct activation of the classical pathway of C by *H. pylori* (I do not know of any precedent here)
- “natural antibody” in NHS that activates via the classical pathway of C (consistent with opinions about serum-sensitivity of other bacteria).

RESPONSE: Thank you for the comment. We were informed by the product supplier that the purchased human serum did not have detectable *H. pylori* specific antibody. In addition, a previous study reported that *H. pylori* is able to trigger classical complement activation even without IgA and IgG that are specific to *H. pylori* (Berstad, 2001). We therefore could imagine that natural antibodies, like IgM, play a role to trigger classical complement activation. To clearly describe this idea, we updated the relevant information in Line 117-118, as “NHS was utilized here as it contains abundant complement components and is able to be activated without *H. pylori* specific antibodies²⁸”.

Reference:

28. Berstad, A. E., Høgåsen, K., Bukholm, G., Moran, A. P. & Brandtzaeg, P. Complement activation directly induced by *Helicobacter pylori*. *Gastroenterology* **120**, 1108–1116 (2001).

REVIEWERS' COMMENTS

Reviewer #1 (Remarks to the Author):

The authors have strengthened the manuscript by incorporating new experimental data including a chronic model of *H. pylori* infection.

Reviewer #3 (Remarks to the Author):

The authors have done a great job in investigating the pH of the mouse antrum and corpus parts in both the WT and the C3-mice (Fig. 1F). The results shows that pH is lower in the corpus region, although this was perhaps the expected. The results might help explain the lower CFUs in the corpus region in the WT mice, since I imagine the bacteria might colonize closer to the epithelium or deeper in the pits/glands to avoid the lower corpus pH. By so doing they might consequentially be more exposed to the complement factors and come out with reduced/lowered infectious load (CFU). The authors might consider reinvestigating their series of slides from Fig 1D and E to also investigate the depth of the infection i.e., the topographic distribution of the GFP *H. pylori* bacterial cells.

Could it possibly be the case that the complement system is active in the lower glandular region to prevent systemic dissemination of the *H. pylori* infection? Have the authors followed mortality and weight gain in the WT vs the C3- mice during the one-year infection study? And also tested for the *H. pylori* (ELISA) antibody responses in the WT vs the C3- mice sera, to understand if there might be more immune responses to the *H. pylori* infection in the C3-mice?

From the results in Fig 1F, I believe we can conclude that the *H. pylori* will be exposed to pH 2.5-3 conditions in the corpus domain. And higher pH in the antrum mucosa. The authors then make use of this result and tested if lower pH affects sera complement activation. In Fig 2A the authors have shown that at pH 7 some 60% of *H. pylori* are killed by complement exposure. In Fig 2B the authors try to use the same system to test activation at the acidic pH 3.5. Although the results suggest that there is still 40 % killing at the pH 3.5, I am in doubt that this is an appropriate way of assaying sera components. From my experience, lowering of the pH of sera samples results in extensive denaturation and precipitation of the sera proteins and the test-results are most likely very difficult to adequately interpret. On the other hand, if it is possible to assay for complement killing efficiency at low pH, I think the authors should also run the tests with the pH values they identified in the corpus tissue i.e., the pH 2.5. Or did this even lower pH fully corrupt the assay system? I imagine the results could show that at corpus low pH, the complement system is totally inactive. But of course, as discussed above, then the *H. pylori* might move closer to the buffered epithelium and deeper into the glands and tissue.

Minor comments,

The illustrated Fig 6 is difficult to understand and would benefit from more explanatory legend-txt. Eg what does the red line holding the C4b illustrate (in the WT) vs the short handle in the susceptible strain? And what do I see in the lower panel with the swimming bugs in the glands? Pls explain.

The Discussion is extensive in volume and would benefit from shortening and more of focus on the complexity of the results and less of general speculation.

Reviewer #4 (Remarks to the Author):

Reviewer #4 (Reply to Authors' RESPONSES)

Major comments

1. The use of chelators to isolate individual complement pathways appeared to indicate that the classical pathway was the only pathway involved in killing *H. pylori*. However, EGTA+Mg²⁺⁺ did

not fully restore survival of organisms in the bactericidal assay, (--compared to survival in EGTA treated sera where the classical pathway is fully blocked); nor would it be expected to because the alternative pathway primarily serves to amplify the classical pathway (not active here because of chelation of Ca^{2++}) although direct activation of the alternative pathway has been suggested by others (*Helicobacter* 1 e, 28-33 (1996)) (zymosan is said to be the only unambiguous direct activator of the alternative pathway). Therefore the alternative pathway cannot be totally ruled out as participating in the role of C in killing *H. pylori*. Furthermore, EGTA alone diminishes free Mg^{2++} but does not eliminate it, possibly permitting expression of some alternative pathway activity. The use of C2 deficient sera (alternative pathway intact) and Factor B (FB) deficient sera or anti-FB reagent (classical pathway intact)—all commercially available, permit a cleaner separation and avoids the effect of chelators on the “health” of the organisms in the bactericidal assays, which can be relevant in the interpretation of results of bactericidal assays. Divalent cations help to maintain integrity of bacterial outer membranes. The killing assays should be repeated using these biologic reagents to corroborate assays that contain chelators. .

RESPONSE: Thank you for the suggestion to use biological reagents to examine *H. pylori* sensitivity to individual complement pathway. As you suggested, we’ve now included data for specific complement component- (C1q, C2, or Factor B) depleted serum and increased the percent of serum as well to gain a cleaner separation of the activation of complement pathways and better determine the sensitivity of *H. pylori* to specific pathway activation. This comment is similar to that of Reviewer 2 #29 and Reviewer 3 #6, so please see the detailed response at Reviewer 2 #29.

Reply: The reply, just above, and the language in the ms. is adequate

2. The bactericidal assays do not indicate the numbers of organism used in the assays; only that the ODs were 0.1; the bactericidal reaction mixtures were diluted before they were plated implying that the reaction mixtures likely had several logs of organisms, yet the displayed data used percentage survival instead of increases/decreases of log amounts—the usual way of describing this type of assay. It is unclear how the authors go from logs of organism in the reaction mixtures to percentage survival in the displayed data! A derivation of these calculations should be described in greater detail for clarification.

RESPONSE: We updated the method for bacterial survival ratio with more clear description, at Line 395 to include the actual bacterial number “...diluted to OD600 of 0.1 (~10⁷ bacteria)” and to describe how we calculated the percentage survival in Line 402-404: “The bacterial survival ratio was calculated by determining the number of colony forming units (CFUs) after exposure to either active NHS or to inactive NHS, and then dividing the numbers in the active serum by those in the inactive.”

Reply: The reply, just above, and the language in the ms. is adequate

Minor comments

1. The first sentence in the Abstract is incorrect. Studies of complement interactions with infectious agents have been studied in other than bloodstream infections; the authors should look these up and modify their statement.

RESPONSE: Thank you for bringing this to our attention. We modified the first sentence of the abstract to “The complement system has long been appreciated for its role in bloodborne infections, but its activities in other places, including the gastrointestinal tract, remain elusive”.

Reply: The reply, just above, and the language in the ms. is adequate

2. Is *H. pylori* infection-free NHS presumed to be free of antibody directed against the organism? What is thought to be the basis for serum-sensitivity of *H. pylori*:

- antibody-free bactericidal activity due to direct activation of the classical pathway of C by *H. pylori* (I do not know of any precedent here)

- “natural antibody” in NHS that activates via the classical pathway of C (consistent with opinions about serum-sensitivity of other bacteria).

RESPONSE: Thank you for the comment. We were informed by the product supplier that the purchased human serum did not have detectable H. pylori specific antibody. In addition, a previous study reported that H. pylori is able to trigger classical complement activation even without IgA and IgG that are specific to H. pylori (Berstad, 2001). We therefore could imagine that natural antibodies, like IgM, play a role to trigger classical complement activation. To clearly describe this idea, we updated the relevant information in Line 117-118, as “NHS was utilized here as it contains abundant complement components and is able to be activated without H. pylori specific antibodies²⁸”.

Reference:

28. Berstad, A. E., Høgåsen, K., Bukholm, G., Moran, A. P. & Brandtzaeg, P. Complement activation directly induced by Helicobacter pylori. Gastroenterology 120, 1108–1116 (2001).

Reply: The reply, just above, and the language in the ms. is adequate

Reviewer #3 (Remarks to the Author):

The authors have done a great job in investigating the pH of the mouse antrum and corpus parts in both the WT and the C3-mice (Fig. 1F). The results shows that pH is lower in the corpus region, although this was perhaps the expected. The results might help explain the lower CFUs in the corpus region in the WT mice, since I imagine the bacteria might colonize closer to the epithelium or deeper in the pits/glands to avoid the lower corpus pH. By so doing they might consequentially be more exposed to the complement factors and come out with reduced/lowered infectious load (CFU). The authors might consider reinvestigating their series of slides from Fig 1D and E to also investigate the depth of the infection i.e., the topographic distribution of the GFP *H. pylori* bacterial cells. Could it possibly be the case that the complement system is active in the lower glandular region to prevent systemic dissemination of the *H. pylori* infection? Have the authors followed mortality and weight gain in the WT vs the C3- mice during the one-year infection study? And also tested for the *H. pylori* (ELISA) antibody responses in the WT vs the C3- mice sera, to understand if there might be more immune responses to the *H. pylori* infection in the C3-mice?

Response: Thank you for this interesting suggestion of investigating the bacterial distribution in the glandular compartments to promote the understanding of complement in various gastric regions. Based on published studies and our own data, we agree with the reviewer's perspective that the complement system restricts *H. pylori* in the deep gland regions. Published work has found that *H. pylori* tends to colonize mainly around the gland base and mid regions, where stem cells are enriched (Sigal et al., 2015). In our manuscript, we show that with the loss of complement, *H. pylori* gland numbers were elevated compared to those in WT mice—in both the percent of occupied gastric glands and the *H. pylori* number per occupied gland. This finding was particularly evident in the corpus region (Fig. 1d, 1e). This information suggests that complement restricts *H. pylori* glandular colonization, especially in the corpus. However, additional questions need to be explored to precisely explain the observation of elevated glandular *H. pylori* number in the complement deficient situation. These include, as the reviewer proposed, alterations of physiology and immune. We believe these are interesting but will require a systematic and detailed analysis that is beyond the scope of this work.

We instead opted to highlight these findings and ideas in the discussion section, and in the proposed model (Fig. 6). In the discussion section, we added “It was noticed that both overall glandular occupancy by *H. pylori* and the numbers per occupied gland were elevated in complement-deficient mice compared to that in WT mice (Fig. 1d, 1e). Our results thus suggest that complement may play an important role in glandular defense, especially in the corpus region, with *H. pylori* utilizing L-lactate to thrive in these glandular regions. Future studies will be important to more precisely elucidate how complement protects gastric glands from microbial infection” (Line 270-275). In the legend for Fig. 6, we updated the description: “Model showing how complement may operate in the stomach to inhibit *H. pylori* colonization. The left panel shows WT *H. pylori*, which is able to colonize the gastric mucosal surface and glands in the face of complement (red circles). The inset details how L-lactate (blue circles) uptake into *H. pylori* blocks the step of C4b surface deposition, rendering *H. pylori* complement resistant. This mechanism of complement resistance is distinct from that used by other bacteria, acting to further block downstream C3 convertase formation and other activities. The right gray panel shows what happen when *H. pylori* is unable to take up L-lactate. In this case, the *H. pylori* is significantly less capable

of destabilizing surface bound C4b, rendering the strain susceptible to complement and less able to colonize the gastric glands and surface”.

Reference:

Sigal, M. *et al.* *Helicobacter pylori* activates and expands LGR5+ stem cells through direct colonization of the gastric glands. *Gastroenterology* **148**, (2015).

From the results in Fig 1F, I believe we can conclude that the *H. pylori* will be exposed to pH 2.5-3 conditions in the corpus domain. And higher pH in the antrum mucosa. The authors then make use of this result and tested if lower pH affects sera complement activation. In Fig 2A the authors have shown that at pH 7 some 60% of *H. pylori* are killed by complement exposure. In Fig 2B the authors try to use the same system to test activation at the acidic pH 3.5. Although the results suggest that there is still 40 % killing at the pH 3.5, I am in doubt that this is an appropriate way of assaying sera components. From my experience, lowering of the pH of sera samples results in extensive denaturation and precipitation of the sera proteins and the test-results are most likely very difficult to adequately interpret. On the other hand, if it is possible to assay for complement killing efficiency at low pH, I think the authors should also run the tests with the pH values they identified in the corpus tissue i.e., the pH 2.5. Or did this even lower pH fully corrupt the assay system? I imagine the results could show that at corpus low pH, the complement system is totally inactive. But of course, as discussed above, then the *H. pylori* might move closer to the buffered epithelium and deeper into the glands and tissue.

Response: We thank the reviewer for raising these questions aimed at more completely understanding the role of complement at acidic conditions. We had similar thoughts, and actually had examined complement killing efficiency at a wide range of pHs. At low pHs, e.g. 2.5, however, we found that cultured *H. pylori* bacteria all died within 1h independent of complement exposure. This finding suggest complement is not necessarily in this condition. Since this is a negative result and has been published previously, we did not show it but just the result at the lowest pH (3.5) for which *H. pylori* did not all die, and that complement played a role in killing *H. pylori* (Fig. 2b). We speculate that the reason that the observed complement killing efficiency was not tremendously reduced from pH of 7 to 3.5 is because *H. pylori* is stressed as well at pH of 3.5, which may enhance complement’s ability to promote *H. pylori* cell lysis. Given the glands are at a more neutral pH than the lumen, this finding is consistent with point 1 above, that complement plays a significant role in glandular defense but it less important in gastric lumen defense.

To incorporate these ideas, we modified our description about the findings of complement killing efficiency at various pH, in the Result section as “These results suggest that complement retains at least some killing ability at low pH” (Line 124-125), and in the Discussion section, as “Our results show that complement retains the ability to kill *H. pylori* even at low pH. Low pH itself is harmful to *H. pylori*, so this outcome may suggest that although complement has reduced activity at low pH⁶¹, there is still enough activity to decrease *H. pylori* numbers”, (Line 287-290).

Reference:

61. Dantas, E. *et al.* Low ph impairs complement-dependent cytotoxicity against IGG-coated target cells. *Oncotarget* (2016).

Minor comments,

The illustrated Fig 6 is difficult to understand and would benefit from more explanatory legend-txt. Eg what does the red line holding the C4b illustrate (in the WT) vs the short handle in the susceptible strain? And what do I see in the lower panel with the swimming bugs in the glands? Pls explain.

Response: Thank you for pointing out that this Figure and description could be clearer. We have modified the figure drawing to more clearly show the lactate transport pathway in both WT and complement-deficient *H. pylori*, to more clearly show the covalent binding between C4b and *H. pylori* surface, and to better depict the bacterial load difference between WT and complement-susceptible *H. pylori* during gastric colonization. We also re-wrote the Figure legend to fully describe what is shown in the diagram and make it more clear, as mentioned in Point-1.

The Discussion is extensive in volume and would benefit from shortening and more of focus on the complexity of the results and less of general speculation.

Response: Thank you for the suggestion to be more focused in the discussion. As suggested, we deleted the paragraph that speculated about the reason that *H. pylori* applies a different anti-complement mechanism (original Line 329-340), and several speculative lines about lactate as a potential signaling molecule (original Line 363-369). We also read the full discussion and trimmed as much as possible.